# Long-term nitrogen fertilization alters microbial respiration sensitivity to temperature and moisture, potentially enhancing soil carbon retention in a boreal Scots pine forest

Boris Ťupek[1], Aleksi Lehtonen[1], Stefano Manzoni[2], Elisa Bruni[3], Petr Baldrian[4], Etienne Richy[4], Bartosz Adamczyk[1], Bertrand Guenet[3], and Raisa Mäkipää[1]

[1]Natural Resources Institute Finland (LUKE), Helsinki, 00790, Finland
[2]Department of Physical Geography and Bolin Centre for Climate Research, Stockholm University, Stockholm, 10691, Sweden.
[3]Laboratoire de Géologie, École Normale Supérieure (ENS), Paris, 75005, France
[4]Laboratory of Environmental Microbiology, Institute of Microbiology of the Czech Academy of Sciences, Prague, 14200, Czech Republic

*Correspondence to*: Boris Ťupek (boris.tupek@luke.fi)

**Abstract.** Nutrient availability affects microbial respiration kinetics and their sensitivities to environmental conditions, thus the soil organic carbon (SOC) stocks. We examined long-term nitrogen (N) addition effects on soil heterotrophic respiration ($R_h$), methane ($CH_4$) oxidation, and nitrous oxide ($N_2O$) emissions in an N-limited boreal Scots pine (*Pinus sylvestris*) forest, in central Finland. Measurements included long-term tree biomass monitoring (1960–2020), soil organic carbon (SOC) in 2023, monthly aboveground litterfall (2021–2023), biweekly $CO_2$, $CH_4$, and $N_2O$ fluxes during the 2021–2023 growing seasons, and quarter-hourly recordings of soil temperature (T) and soil water content (SWC) in both control and N-fertilized plots. We assessed mean greenhouse gas (GHG) flux differences and $R_h$ dependence on T and SWC using polynomial and non-linear regression models.

Tree biomass, litterfall and SOC increased with long-term N fertilization. However, N fertilization also significantly increased mean $R_h$, reduced $CH_4$ oxidation slightly, and modestly raised $N_2O$ emissions. SOC-normalized $R_h$ ($R_{h/SOC}$) did not significantly differ between treatments, yet relationships between $R_{h/SOC}$ and T and SWC diverged with fertilization. In control plots, $R_{h/SOC}$ peaked at 15.8 °C and at 16.8 °C in N-fertilized plots. Under N fertilization, $R_{h/SOC}$ was weakly SWC-dependent, contrasting with a distinct humped SWC response enhancing annual $R_{h/SOC}$ in control plots. Annually, N-fertilized plots respired 10.3% of SOC (± 0.3 standard error (SE)), compared to 12.2% (± 0.5 SE) in controls, suggesting N fertilization promoted SOC retention. Consequently, N fertilization reduced average annual net $CO_2$ emissions by 345.4 (± 73.6 SE) g $CO_2$ m$^{-2}$ year$^{-1}$, while combined effects on $CH_4$ and $N_2O$ fluxes and the production energy of N fertilizer contributed annually a minor $CO_2$-equivalent increase of 17.7 (± 0.5 SE) g $CO_2$-eq m$^{-2}$ year$^{-1}$.

In conclusion, long-term N fertilization in boreal forests could reduce global warming potential of soil GHG emissions, mainly by slowing $R_{h/SOC}$, and altering its responses to T and SWC, thereby enhancing SOC sequestration in addition to the increased tree biomass carbon sink.

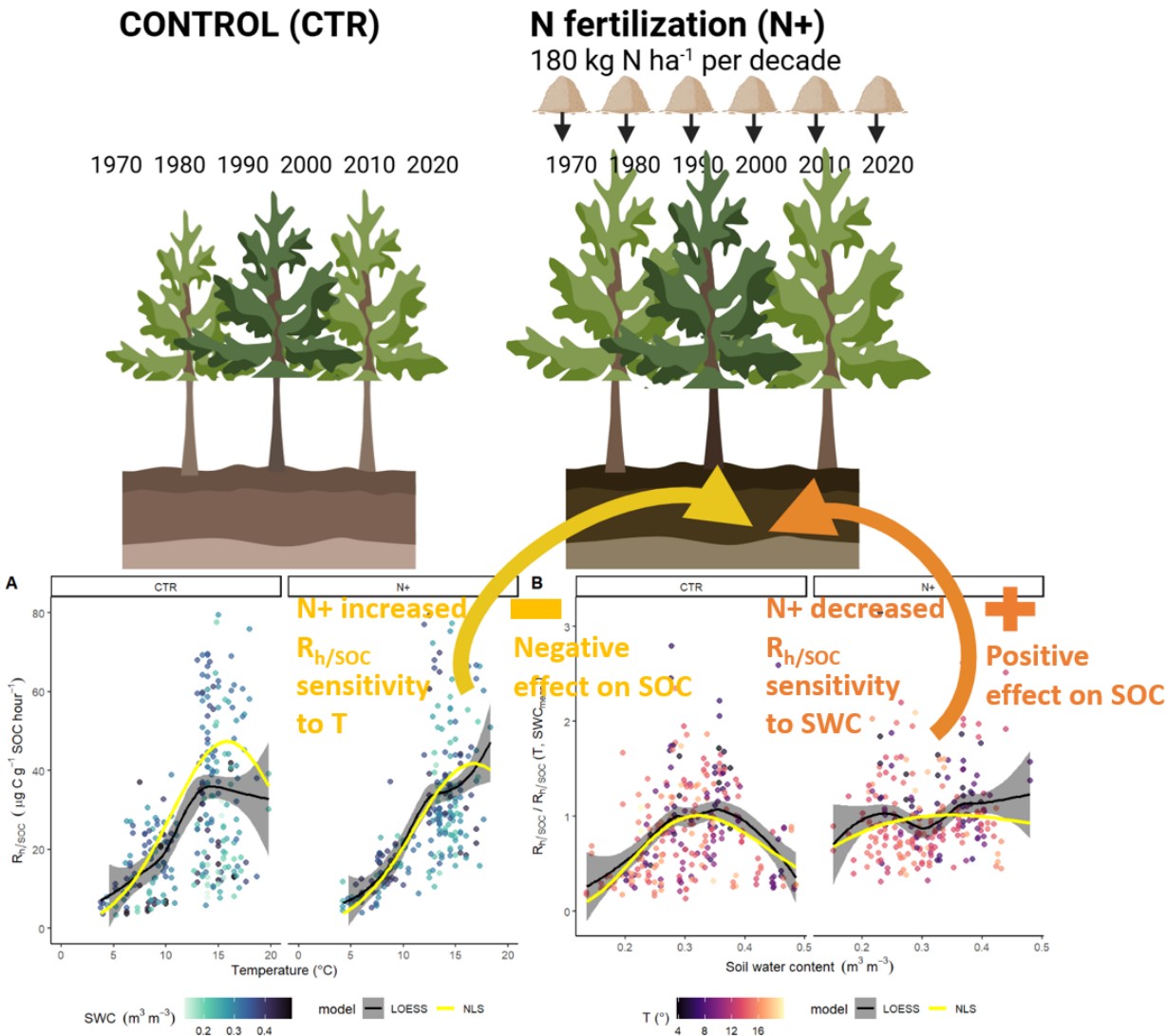

## 1 Introduction

Vegetation growth in boreal forests is primarily constrained by temperature (Jarvis and Linder, 2000) and soil nutrient availability, particularly nitrogen (N) (Näsholm et al., 1998; Högberg et al., 2017). Atmospheric N deposition or fertilization can enhance tree biomass growth (Lupi et al., 2013; Saarsalmi and Mälkönen, 2001; Sponseller et al., 2016) and increase soil carbon (C) sequestration by promoting productivity and litter inputs while reducing decomposition rates (Janssens et al., 2010; Marshall et al., 2021; Smolander et al., 1994). This increased C storage in both tree biomass and soil after N

fertilization could be seen as a positive feedback effect on ecosystem C balance in Northern forests (Hyvönen et al., 2008; Mäkipää et al., 2023). However, the effects of N fertilization on organic matter (OM) decomposition and the net balance of greenhouse gas (GHG) emissions ($CO_2$, $CH_4$, $N_2O$) are less well understood and equally critical for assessing the forest C balance and its global warming potential. N fertilization may reduce soil $CO_2$ emissions (Janssens et al., 2010) due to enhanced microbial carbon use efficiency (CUE) (Manzoni et al., 2012b, 2017) and decreased need for N mining from organic matter (Craine et al., 2007). It may also increase $N_2O$ emissions due to greater soil N availability (Högberg et al., 2017; Öquist et al., 2024) and potentially alter $CH_4$ uptake by either increasing N availability for $CH_4$ oxidizing microbes or by competing with $NH_4$ for reduction (Öquist et al., 2024). Because these soil processes could alter the forest C balance, offset the enhanced tree C sink, potentially converting the ecosystem into a net C source. Evaluating the feedback of N fertilization on forest climate mitigation potential requires consideration of impacts on both tree growth and OM decomposition. Moreover, full accounting of GHG emissions should include emissions associated with N fertilizer production (Osorio-Tejada et al., 2022).

The soil C balance in boreal forests, which is a dynamic balance between C input from litterfall and $CO_2$ emissions from OM decomposition, is influenced by temperature (T), soil water content (SWC), nutrient status, and vegetation composition (Deluca and Boisvenue, 2012)—all factors sensitive to forest management (Mäkipää et al., 2023; Mayer et al., 2020). For example, N fertilization enhances soil N availability, promoting plant growth and litterfall (C input) while potentially reducing OM decomposition due to increased CUE of N-limited microbial decomposers (Manzoni et al., 2017). These effects, alongside T and SWC controls, can be integrated into soil C models (Zhang et al., 2018). Moreover, changes in microbial community structure (e.g., activity, CUE, and biodiversity; Khurana et al., 2023) induced by fertilization can affect decomposition dynamics and influence soil microbial respiration dependencies on T and SWC. For example, shifts in respiration responses to temperature due to N fertilization may attenuate $CO_2$ emissions under warming scenarios (Chen et al., 2024; Wei et al., 2017). Although the effects of N addition on moisture dependency remain understudied, interactions between T and SWC are critical for forecasting respiration responses (Pallandt et al., 2022; Sierra et al., 2017, 2015).

Empirically derived relationships between soil respiration and T and SWC are widely used in soil C models to adjust decomposition rate constants (Luo et al., 2016), yet differences in SWC responses (Sierra et al., 2015) contribute to projection uncertainties (Falloon et al., 2011). Boreal forest soils with higher nutrient levels and water availability often have underestimated SOC stocks in model projections (Dalsgaard et al., 2016; Tupek et al., 2016). Moreover, SWC response curves vary with soil properties like porosity, clay content, and OM fraction (Moyano et al., 2013, 2012) and may also be influenced by soil N status. Given the significant spatial variability in SOC within forest sites (Muukkonen et al., 2009) and the measurement uncertainty over time (Ortiz et al., 2013), assessing changes in the T and SWC dependencies of soil $CO_2$ emissions after long-term N fertilization and applying them over multiple years could clarify the SOC sink/source dynamics.

In southern boreal region's Scots pine forests on well-drained, often N-poor mineral soils, soil $CO_2$ emissions range from 1 to 3 kg $CO_2$ $m^{-2}$ $year^{-1}$, accounting for 70–91% of total ecosystem respiration (Tupek et al., 2008; Uri et al., 2022) and its global warming potential (GWP). Although $CH_4$ and $N_2O$ have higher GWP than $CO_2$ (27 and 273 times over a 100-year

horizon, respectively; IPCC (2023)), the soil generally acts as a small $CH_4$ sink, and $N_2O$ emissions are negligible in these N-limited soils (Machacova et al., 2016; Matson et al., 2009; Pihlatie et al., 2007; Siljanen et al., 2020; Tupek et al., 2015).

In this study, we investigated the effects of long-term N fertilization on soil $CO_2$, $CH_4$, and $N_2O$ fluxes and SOC stocks in a boreal Scots pine forest. We hypothesized that (i) increased soil nitrogen availability would enhance soil organic carbon (SOC) accumulation and heterotrophic respiration ($R_h$) due to greater biomass growth and litter inputs, while SOC-normalized $R_h$ ($R_{h/SOC}$) would decline due to reduced microbial nitrogen mining; and (ii) nitrogen fertilization would alter $CH_4$ oxidation and increase $N_2O$ emissions compared to N-limited soils, reflecting shifts in microbial activity and substrate availability.

## 2 Methods

### 2.1 Study site and N fertilization

The Karstula forest study site is in central Finland (62°54'43.343"N, 24°34'16.021"E) (Fig. 1) and is dominated by Pinus sylvestris (Scots pine) with an understory comprising *Vaccinium myrtillus*, *V. vitis-idaea*, *Empetrum nigrum*, *Calluna vulgaris*, and various boreal mosses and lichens. Established on a low-fertility sandy podzol, the site corresponds to the Calluna (CT) and Vaccinium vitis-idaea (VT) types in the Finnish classification system (Cajander, 1949). Nitrogen (N) fertilization has been applied here since 1960, with 180 kg N ha⁻¹ potassium nitrate applied every decade until 2020.

The stand underwent thinning in 1990 and 2015. To maintain comparable management across fertilization treatments, both CTR and N-fertilized (N+) plots were thinned in 1990 with similar intensity (~20%), and again in 2015 with nearly identical intensity, reducing basal area by 36.7% (CTR) and 40.1% (N+), following the Finnish silvicultural guidelines (Tapio, www.tapio.fi).

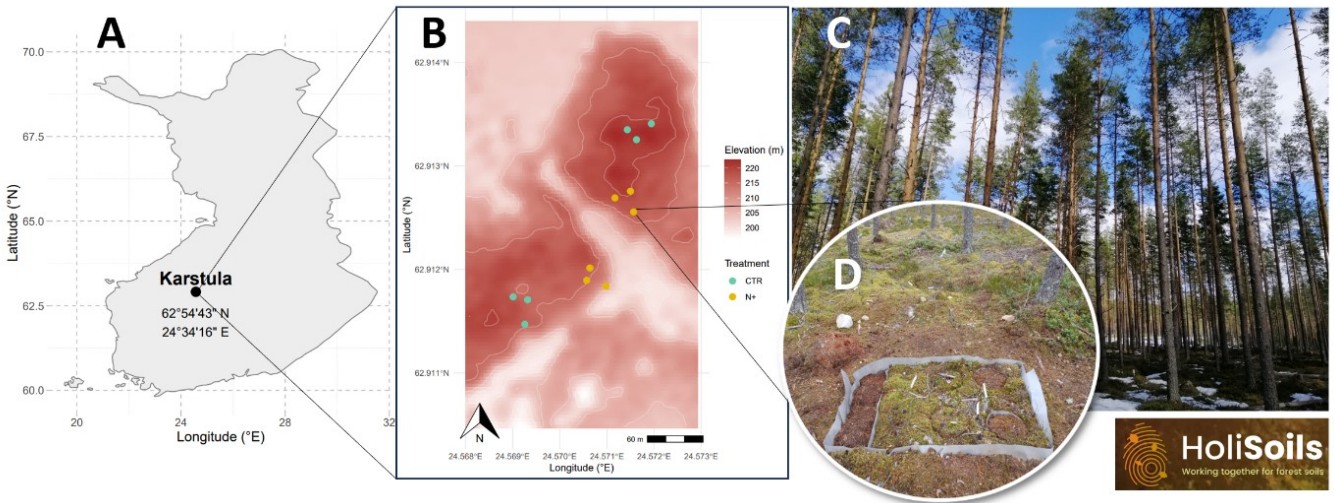

**Figure 1: Geographical location of the Karstula forest study site in Finland (A); topographical variation of the study site and the location of treatment (control CTR and N-fertilized N+) plots (B); photograph of the forest stand (C); and one of six 2 x 1 m forest floor plot groups, each with four subplots used for measuring soil greenhouse gases, soil organic C, and soil temperature and moisture following the installation of a root-exclusion fabric (D).**

## 2.2 Field measurements

### 2.2.1 Tree inventory and litterfall

Measurements of tree diameter (at 1.3 m height), total height, and crown base height have been conducted at 10-year intervals from 1960 to 2010 and every 5 years thereafter. The breast-height diameter (d1.3) of all trees has been measured using a caliper (±1 mm precision) once per decade since 1960, as well as after the 2015 thinning. Additionally, in a permanent subset of trees representing various size categories, tree height and crown base height have been recorded using a hypsometer (precision ~0.5–1 m). Mortality and thinning-related removals were recorded, and tree biomass was calculated using biomass expansion models (Repola, 2009). Litterfall rates were estimated using compartment-specific turnover rates (Lehtonen et al., 2016). From May 2021 to October 2023, litter (needles, twigs, cones) was collected monthly during the growing season using 0.8 m mesh collectors and subsequently sorted and weighed.

### 2.2.2 Soil organic carbon stock (SOC)

Soil sampling was performed in June 2023 in control and N-fertilized plots (n=6 each) using a 58 mm diameter corer. Samples were stratified by layer, separating humus from mineral soil, which was sampled in 10 cm increments to a depth of 30 cm. Samples from each layer were composited across two subsites with differing rock content. Samples were dried, weighed, and sieved, and C and N contents were analyzed using dry combustion (LECO TruMac CN, LECO Corporation, St. Joseph, MI, USA). Stoniness was assessed in the field using rod penetration (Eriksson and Holmgren, 1996) and corrected for rock fragment content following Poeplau et al. (2017).

### 2.2.2 Soil greenhouse gas (GHG) fluxes, temperature, and moisture

Soil GHG fluxes ($CO_2$, $CH_4$, and $N_2O$) were measured biweekly during the growing seasons of 2021-2023 (spanning from 20th May to 16th August in 2021, 5th May to 3rd November in 2022, and 10th May to 10th October in 2023). In May 2021, three $1 \times 2$ m trenched areas were established per treatment. Each trench was lined with water-permeable geotextile to prevent root ingrowth, thereby isolating heterotrophic respiration ($R_h$) from autotrophic sources (Tupek et al., 2019). Measurements were taken from 12 plots (six per treatment; Fig. 1b). Two plot pairs ($2 \times 706$ cm²) were used to account for local heterogeneity in soil and microtopography within each trenched area (1 m²), while three trenched areas per treatment were used to capture spatial heterogeneity at the site level (Fig. 1d). Plots in each pair were located 30 cm apart (Fig. 1c) and CTR and N+ pairs were on average 122 m apart (Fig. 1b).

Gas fluxes were measured using a non-transparent 21.7 L dynamic chamber (30 cm in diameter and height) equipped with a fan and connected to a LI-COR LI-7810 $CH_4/CO_2/H_2O$ or LI-7820 $N_2O/H_2O$ trace gas analyzer (LICOR, Lincoln, NE, USA). Gas concentrations were recorded every second for 3 minutes, and linearity was monitored visually during the measurements to accept only fluxes with increasing trends in $CO_2$ concentration evolution. Fluxes were calculated from the stable portion of the data (Zhao, 2019). $R_h$ values (g $CO_2$ $m^{-2}$ $h^{-1}$) were normalized to SOC content and expressed as a C fraction of SOC per hour (µg C $g^{-1}$ SOC $h^{-1}$). The $CH_4$ and $N_2O$ concentrations were also measured during 3 min intervals with 5 second averaging at the 0.25 ppb precision for $CH_4$ and 0.20 ppb precision for $N_2O$. The minimum detectable flux of measurements estimated using the formula by Parkin et al., (2012) was 0.0238 µg $m^{-2}$ $h^{-1}$ for $CH_4$ and 0.0524 µg $m^{-2}$ $h^{-1}$ for $N_2O$.

Soil temperature (T) and volumetric soil moisture (SWC) at 5 cm depth were continuously monitored with Soil Scout Oy sensors, recording data at 15 min intervals since June 2021. T and SWC were matched with flux data by timestamp.

**2.3 Data analysis**

All data analyses and visualizations were conducted using R software (R Core Team, 2023). The full data set and the R code for producing the analysis and results described below are available on Zenodo (Tupek et al. 2024, Tupek B., 2024). A one-way ANOVA was employed to test the effect of N fertilization on greenhouse gas (GHG) fluxes. Since the data were collected at relatively low temporal frequency (biweekly), the degree of temporal autocorrelation was substantially lower than in high-frequency (e.g., hourly) automated measurements. Therefore, no additional correction for autocorrelation was applied. The independence assumption of ANOVA was considered reasonably met under these conditions.

Two regression approaches were used to characterize the dependency of $R_{h/soc}$ on T and SWC: (i) local polynomial regression (LOESS) to assess the functional form of $R_{h/soc}$ dependencies on combined T and SWC ($R_{h/soc}$(T, SWC)) separately for the N-fertilized (N+) and control (CTR) plots; and (ii) nonlinear least squares (NLS) regression, guided by LOESS to identify suitable mathematical forms. The LOESS and NLS models for $R_{h/soc}$ dependency on SWC alone were compared using $R_{h/soc}$ ratios normalized by $R_{h/soc}$(T, $SWC_{mean}$).

In approach (ii), the combined T and SWC dependency of $R_{h/soc}$ was modeled by multiplying a Gaussian T function as described in Tuomi et al. (2008) with a Ricker function for SWC (Bolker, 2008) (Eq. 1):

$$R_{h/soc}(T, SWC) = e^{(\beta_1 T + \beta_2 T^2)}(a\ SWC\ e^{(-b\ SWC)})^c ,\hspace{2cm}(1)$$

where $\beta_1$ and $\beta_2$ are parameters controlling the exponential T response, parameter a determines the initial slope and rescales the whole function, exponent b describes the post-optimal decline, and exponent c modulates the sharpness of the peak of the SWC response.

Model performance was assessed using proportion of explained variance ($R^2$), root mean square error (RMSE), mean bias error (MBE), and mean absolute error (MAE) derived from model residuals. Model robustness was further evaluated with 10-fold cross-validation (Kuhn, 2008).

Once the parameters of the T and SWC response were determined, the NLS regression was used to extrapolate $R_{h/SOC}$ to continuous hourly data and to upscale $R_{h/SOC}$ to the annual level. Annual $CH_4$ and $N_2O$ fluxes were estimated by scaling the treatment-specific mean hourly flux values without considering T and SWC effects. The global warming potential (GWP) was calculated using the AR6 GWP-100 values (27 for $CH_4$ and 273 for $N_2O$) (IPCC, 2023). As flux data were unavailable for the November–March period, the $CH_4$ and $N_2O$ annual estimates are limited to the extrapolating the conditions of Apr–Oct, during which fluxes are generally higher.

The emissions associated with fertilizer production were accounted for according to Osorio-Tejada et al. (2022). We estimated the $CO_2$ emissions associated with six nitrogen fertilization events, which occurred once per decade between 1960 and 2020. The applied nitrogen fertilization rate was 180 kg N ha$^{-1}$ per event. Converting this to ammonia ($NH_3$) using the molecular weight ratio of $NH_3$ to N (17.031/14.007) resulted in an estimated 218.86 kg $NH_3$ ha$^{-1}$ per fertilization event. Given an emission factor of 2.96 kg $CO_2$ per kg $NH_3$, this corresponds to 647.93 kg $CO_2$ ha$^{-1}$ per event. Over six fertilization events spanning 60 years, the annualized $CO_2$ emission was calculated as 64.79 kg $CO_2$ ha$^{-1}$ yr$^{-1}$, equivalent to approximately 6.5 g $CO_2$ m$^{-2}$ yr$^{-1}$.

## 3 Results

### 3.1 N fertilization enhanced tree biomass, litterfall, and SOC

N fertilization led to increased tree stand biomass and litterfall in N+ compared to CTR plots. Despite reductions following thinning events, tree biomass was highest in 2014 for both treatments (9 kg C m$^{-2}$ in N+ and 7 kg C m$^{-2}$ in CTR), decreasing to 6 and 5 kg C m$^{-2}$, respectively, by 2020 due to thinning in 2015 (Fig. 2a). This thinning led to peak litter input in 2015 (1.5 kg C m$^{-2}$ in N+ and 1 kg C m$^{-2}$ in CTR), which then stabilized around 0.6 and 0.5 kg C m$^{-2}$ due to the presence of fewer trees (Fig. 2a). Litter fraction accounted for 16% of N+ and 14% of CTR biomass in 2015, falling to 10% for both by 2020. Monthly litterfall, including needles, branches, and cones, was significantly higher in N+ (25.1 g m$^{-2}$ month$^{-1}$) than in CTR

(14.3 g m⁻² month⁻¹) plots from 2021 to 2023 (Fig. 2b). SOC tended to be higher under N fertilization, from 4.9 kg C m⁻² in CTR to 5.6 kg C m⁻² in N+, but the difference was not statistically significant. (Fig. 2c).

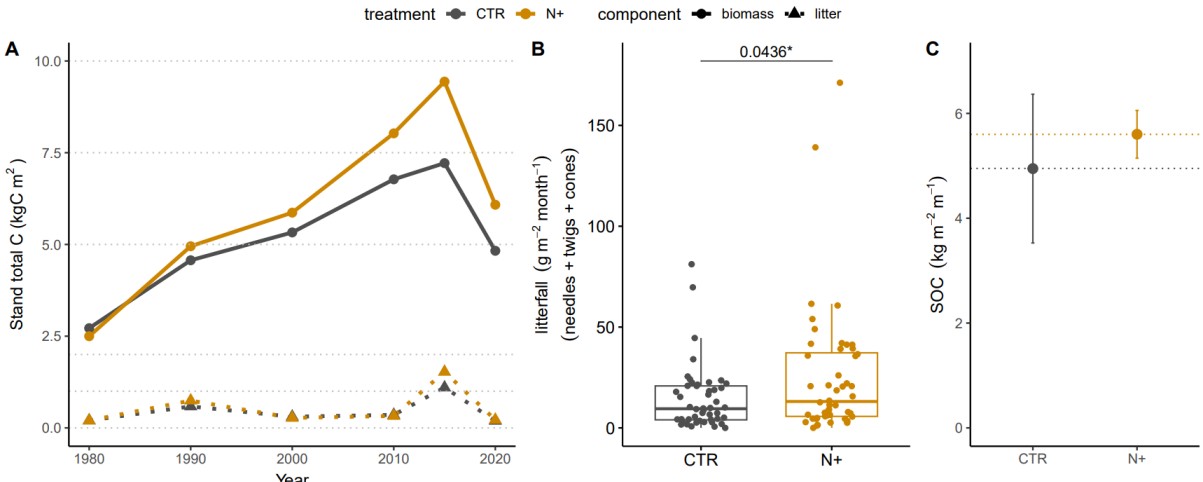

**Figure 2: Biomass, litterfall, and SOC in control (CTR) and N-fertilized (N⁺) stands. (A) Estimated tree biomass and litterfall from 1980 to 2020 forest tree stands inventory measurements. (B) Monthly litterfall from July 2021 to October 2023 (box plot shows median, quartiles, and outliers). (C) SOC stock over 1 m depth in 2023 (error bars indicate minimum and maximum values across replicates).**

### 3.2 N fertilization effects on mean CO₂, CH₄, and N₂O fluxes but not on SOC-normalized CO₂

Pairwise ANOVA showed that mean annual soil microbial respiration $R_h$ (g CO₂ m⁻² h⁻¹) was significantly higher in N+ (0.58 ± 0.01 standard error, SE) than in CTR plots (0.46 ± 0.01 SE) (F-value 15.96, degrees of freedom 449, p-value 8.92e-05) (Fig. 3a). However, $R_h$ normalized by SOC (µg C g⁻¹ SOC h⁻¹) did not differ significantly between N+ (28.3 ± 1.1 SE) and CTR plots (28.6 ± 1.1 SE) (Fig. 3b).

CH₄ oxidation was slower in N+ (-0.14 ± 0.002 SE mg CH₄ m⁻² h⁻¹) than in CTR (-0.18 ± 0.002 SE mg CH₄ m⁻² h⁻¹) (Fig. 3c), with annual CH₄ oxidation rates of -1.58 g CH₄ m⁻² y⁻¹ in CTR and -1.21 g CH₄ m⁻² y⁻¹ in N⁺ plots. Mean net N₂O exchange was significantly lower than zero in CTR (-0.25 ± 0.09 SE µg N₂O m⁻² h⁻¹), while in N+ it was positive (0.22 ± 0.06 SE µg N₂O m⁻² h⁻¹), resulting in a mean annual difference of 4.17 mg N₂O m⁻² y⁻¹ between treatments (Fig. 3d). The method detection limits were smaller than SE of mean CH₄ and N₂O fluxes.

Average T at 5 cm depth was higher in CTR (12.6 ± 0.17 SE °C) than in N+ (12.0 ± 0.16 SE °C) (Fig. 3e), while SWC at 5 cm depth (0.31 m³ m⁻³) did not differ significantly between treatments (Fig. 3f). Mean annual T was 5.92 ± 0.18 SE °C in

CTR and 5.83 ± 0.17 SE °C in N⁺, with an annual SWC of 0.31 ± 0.002 SE m³ m⁻³ for both (Fig. S1). Soil T increased

rapidly after snowmelt in April, with cooler summer conditions in 2022 than in 2021 and 2023. SWC ranged from 0.07 to 0.42 m³ m⁻³, dropping below 0.2 m³ m⁻³ during drought conditions in summer 2021 (Fig. S1, S2). $R_h$ showed sensitivity to T and SWC, rising with warmer conditions and declining in dry periods, then recovering after rewetting events (Fig. S2). However, this pattern was more pronounced in CTR than in N+ plots. Part of the variation in soil moisture between CTR and N+ plots (located on average 122 m apart) could be attributed to the measured topsoil humus layer being affected by

microscale variations of vertical and lateral water flows due to microtopography (Fig. 1b).

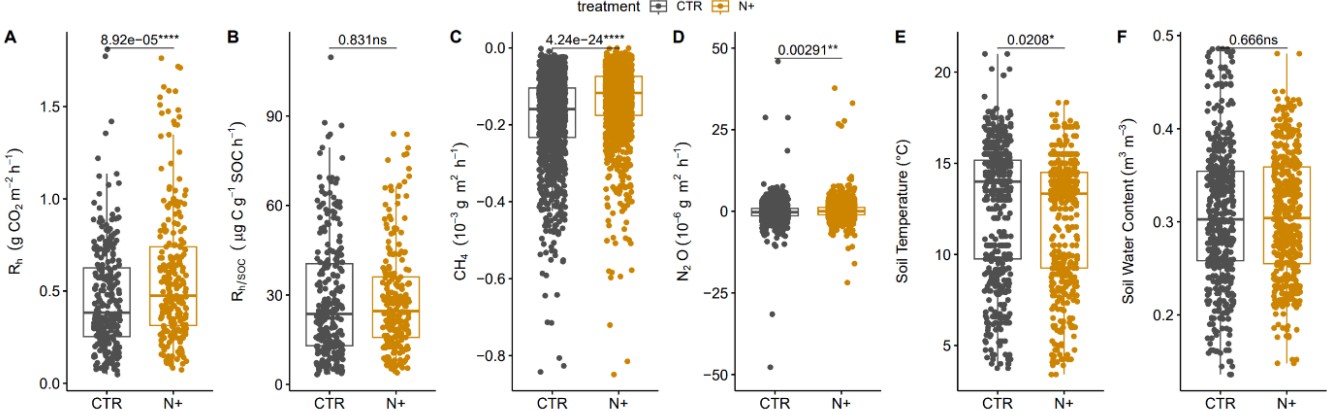

**Figure 3: Soil (A) heterotrophic respiration ($R_h$, g CO₂ m⁻² h⁻¹), (B) $R_h$ normalized by SOC (μg C g⁻¹ SOC h⁻¹), (C) net CH₄ flux (mg CH₄ m⁻² h⁻¹), (D) net N₂O flux (μg N₂O m⁻² h⁻¹), (E) soil temperature (T, °C), and (F) soil volumetric water content (SWC, m³**

**m⁻³) for N⁺ and CTR plots in 2021, 2022, and 2023 field campaigns.**

### 3.3 N fertilization altered $R_{h/SOC}$ dependencies on T and SWC

LOESS and NLS regression models showed similar $R_{h/SOC}$ dependencies on T and SWC (Fig. 4a, 4b). In CTR and N+, NLS models indicated a T optimum at 15.8 °C and 16.8 °C, respectively, above which decomposition was limited by dry soil

conditions. Thus $R_{h/SOC}$ in CTR at T below the optimum rose more steeply compared to N+ plots (Fig. 4a).

The $R_{h/SOC}$ revealed an SWC optimum in CTR, while in N+ plots the $R_{h/SOC}$ - SWC dependency was less pronounced (Fig. 4b). The $R_{h/SOC}$ was maximized at SWC = 0.32 m³ m⁻³ in CTR and at SWC = 0.35 m³ m⁻³ in N+, and in CTR it declined more steeply under both drier and wetter conditions than in N+.

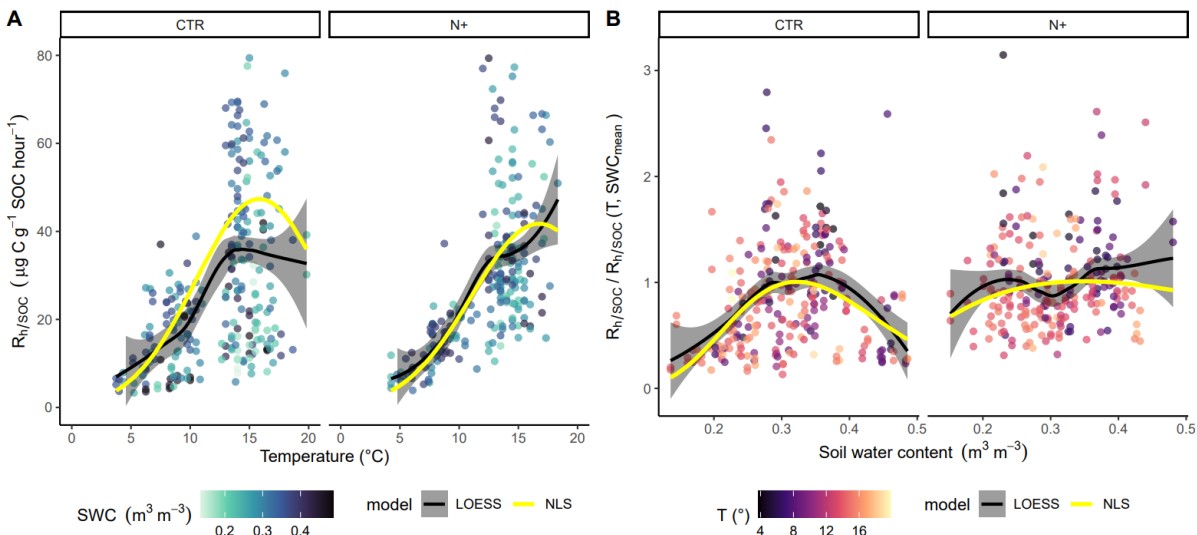

**Figure 4: (A)** Dependence of soil microbial respiration normalized by soil organic carbon ($R_{h/SOC}$, µg C g$^{-1}$ SOC h$^{-1}$) on soil temperature at 5 cm depth (T, °C). **(B)** Ratio of measured $R_{h/SOC}$ to modeled $R_{h/SOC}$(T, SWC$_{mean}$) as a function of volumetric water content (SWC, m³ m$^{-3}$) at 5 cm depth. Panels display results separately for control (CTR) and N-fertilized (N+) plots. Shading of turquoise points in (A) reflects varying SWC, while shading of red points in (B) corresponds to variation in T. Black lines indicate local polynomial regression (LOESS) fits with gray ribbons showing 95% confidence intervals; yellow lines represent nonlinear least square (NLS) regression model fits. The NLS lines in (A) are modeled as $R_{h/SOC}$(T, SWC$_{mean}$) and in (B) as $R_{h/SOC}$(T, SWC)/ $R_{h/SOC}$(T, SWC$_{mean}$).

Parameters and fit statistics for the NLS regression model are provided in Table 1 and Table 2. In CTR, the Ricker power parameter $c$ significantly differed from one, indicating suppressed respiration in non-optimal SWC conditions. The model parameters describing functional dependencies on soil moisture were statistically different from zero for CTR but not for N+ (Table 1). While not statistically different from zero in N+ plots, the c value near 1 suggests a relatively flat response of $R_{h/SOC}$ to SWC. In contrast, a high c value ($\approx$ 8, p < 0.001) in CTR plots indicates a peaked moisture response, supporting a stronger effect of moisture limitation on decomposition under ambient conditions (Table 1). Neither the CTR nor N+ models showed bias (Table 2 and S1). NLS model fit metrics showed that R$^2$, RMSE, MBE, and MAE values were comparable between CTR and N+ models (Table 2). RMSE and MAE for NLS models fell within the range of standard deviations from 10-fold cross-validation (Table S1). The model accuracy was generally higher in N+ than in CTR plots (Table 2).

**Table 1: Parameter estimates with standard errors (SE) and p-values for combined temperature and SWC models (Eq. 1: $\beta_1$, $\beta_2$, a, b, and c).**

| Treatment | Parameter | Estimate | SE | p-value |
|---|---|---|---|---|
| CTR | $\beta_1$ | 0.545 | 0.101 | <0.001 |
| | $\beta_2$ | -0.017 | 0.004 | <0.001 |
| | a | 7.967 | 0.703 | <0.001 |
| | b | 3.101 | 0.073 | <0.001 |
| | c | 8.045 | 1.347 | <0.001 |
| N+ | $\beta_1$ | 0.515 | 0.105 | <0.001 |
| | $\beta_2$ | -0.015 | 0.004 | <0.001 |
| | a | 5.317 | 3.250 | 0.103 |
| | b | 2.871 | 0.432 | <0.001 |
| | c | 1.500 | 1.063 | 0.160 |

**Table 2: Goodness-of-fit statistics for NLS models based on combined temperature and moisture (Eq. 1): proportion of explained variance ($R^2$), root mean square error (RMSE), mean bias error (MBE), and mean absolute error (MAE). RMSE, MBE and MAE in $\mu g\ C\ g^{-1}\ SOC\ h^{-1}$.**

| Treatment | $R^2$ | RMSE | MBE | MAE |
|---|---|---|---|---|
| | | $\mu g\ C\ g^{-1}\ SOC\ h^{-1}$ | | |
| CTR | 0.41 | 15.55 | -0.33 | 11.42 |
| N+ | 0.40 | 13.36 | -0.48 | 9.28 |

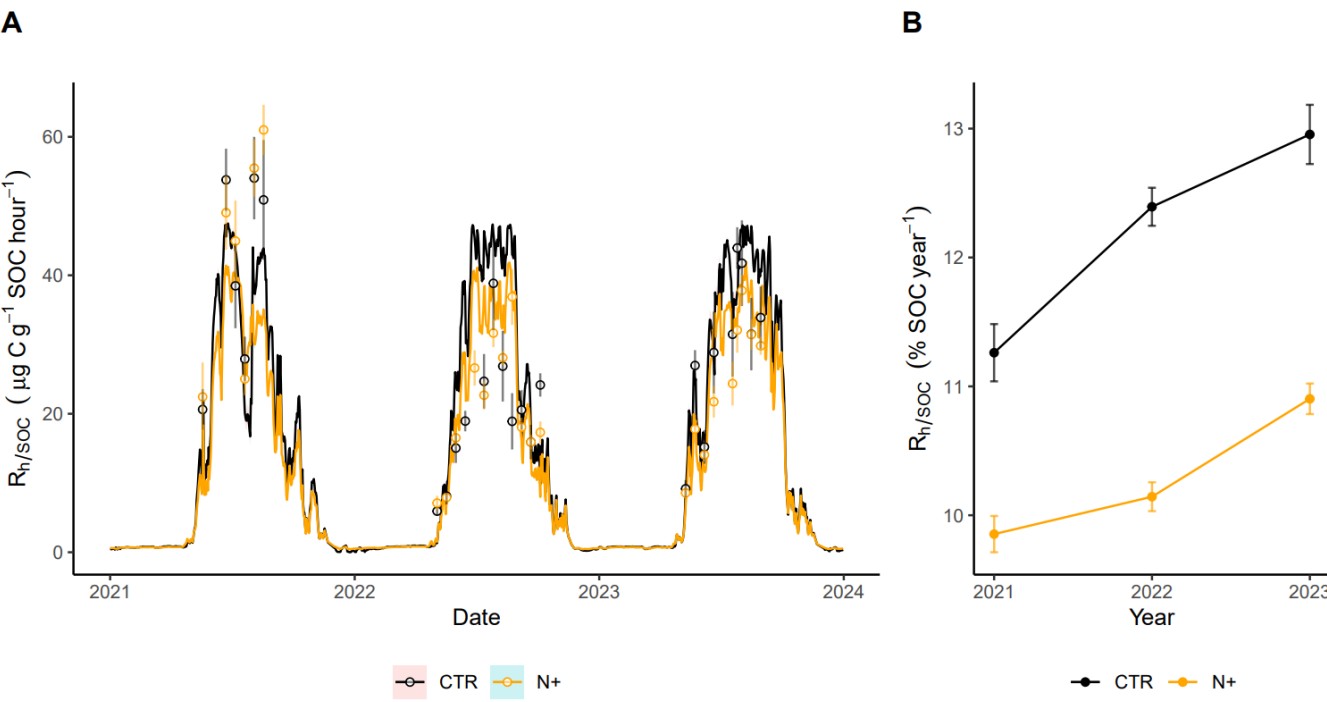

**Figure 5: A) Time series of daily mean $R_{h/SOC}$ (µg C g⁻¹ SOC h⁻¹) in CTR and N+ plots, with measurements shown as points (error bars indicate SE) and model estimates as lines (ribbons indicate SE). B) Annual $R_{h/SOC}$ (% SOC per year) in CTR and N+ plots.**

### 3.4 Seasonal and annual differences in $R_{h/SOC}$ between CTR and N+ plots

During the winter, daily model predictions of $R_{h/SOC}$ remained consistent across CTR and N+ treatments, with little variation due to low soil temperatures (Fig. 5a). However, in the summer, with temperatures above 5°C, $R_{h/SOC}$ modeled with the function $R_{h/SOC}$(T, SWC) displayed marked differences between CTR plots and N+ plots. In CTR plots, the modeled $R_{h/SOC}$ values were generally higher than in the N+ plots, except during a brief drought period in 2021, when modeled $R_{h/SOC}$ values were lower..

### 3.5 Annual GWP reduction in relation to N addition

As a result of the generally lower and less variable daily $R_{h/SOC}$ in N+ plots, the annual $R_{h/SOC}$ (expressed as %  SOC respired per year) in N+ was also consistently lower than in CTR (Fig. 5b). Annual $R_{h/SOC}$ rates ranged from 9.85 (±0.14 SE) to 12.95 (±0.23 SE) % and increased during the 2021–2023 period (Fig. 5b). On average, $R_{h/SOC}$(T, SWC) values were  1.90 (±0.41 SE) % higher in control (CTR) plots compared to N-fertilized (N+) plots.. This result suggests that, despite increased litter inputs in N+ plots due to enhanced tree growth, the relative decomposition rate per unit SOC remained unchanged or declined in fertilized plots, potentially favouring greater SOC retention.The difference in modelled $R_h$ (calculated as

$R_{h/SOC}$(T, SWC) × SOC) between CTR and N+ plots corresponds to a reduction of -345.4 (±73.6 SE) g $CO_2$ $m^{-2}$ $yr^{-1}$ in heterotrophic $CO_2$ emissions (Table 3). This potential reduction in $CO_2$ emissions outweighed the GWP associated with increased $N_2O$ emissions (1.1 ± 0.1 g $CO_2$-eq $m^{-2}$ $yr^{-1}$), reduced $CH_4$ uptake (10.1 ± 0.5 g $CO_2$-eq $m^{-2}$ $yr^{-1}$) and fertilizer production emissions (6.5 g $CO_2$-eq $m^{-2}$ $yr^{-1}$). Overall, the net GWP balance suggests an annual reduction of –327.6 (±73.6 SE) g $CO_2$-eq $m^{-2}$ $yr^{-1}$ attributable to N fertilization.

**Table 3: Annual global warming potential (GWP) reduction by long-term N fertilization in boreal Scots pine forest with contribution of individual greenhouse gas (GHG) fluxes (microbial respiration normalized by soil organic carbon stock $R_{h/SOC}$, $CH_4$ net oxidation, and $N_2O$ flux net exchange) evaluated as a difference between control (CTR) and N fertilized plots (N+). Minus values indicate net GWP reduction. The AR6 GWP-100 values 27 for $CH_4$ and 273 for $N_2O$ were used for calculation of $CO_2$-equivalents (ICCP, 2023). SE values were calculated considering variations across replicates and years together.**

| Treatment | $R_{h/SOC}$ (%) | | GWP-$CO_2$ (g $CO_2$ $m^{-2}$ $y^{-1}$) | | $CH_4$ (g $CH_4$ $m^{-2}$ $y^{-1}$) | | GWP-$CH_4$ (g $CO_2$-eq $m^{-2}$ $y^{-1}$) | | $N_2O$ (mg $N_2O$ $m^{-2}$ $y^{-1}$) | | GWP-$N_2O$ (g $CO_2$-eq $m^{-2}$ $y^{-1}$) | | GWP-GHG (g $CO_2$-eq $m^{-2}$ $y^{-1}$) | |
|---|---|---|---|---|---|---|---|---|---|---|---|---|---|---|
| | mean | SE | mean | SE | mean | SE | mean | SE | mean | SE | mean | SE | mean | SE |
| **CTR** | 12.2 | 0.5 | 2214.9 | 90.4 | -1.6 | 0.0 | -42.9 | 0.5 | -2.2 | 0.8 | -0.6 | 0.2 | 2171.4 | 90.4 |
| **N+** | 10.3 | 0.3 | 1869.5 | 56.8 | -1.2 | 0.0 | -32.8 | 0.5 | 1.9 | 0.6 | 0.5 | 0.2 | 1837.3 | 56.8 |
| **Difference** | -1.9 | 0.4 | **-345.4** | **73.6** | 0.4 | 0.0 | **10.1** | **0.5** | 4.2 | 0.7 | **1.1** | **0.2** | **-334.1** | **73.6** |

## 4 Discussion

Our results show that nitrogen (N) fertilization significantly increased tree stand biomass and litterfall in N+ plots compared to control (CTR) plots (Fig. 2a), aligning with previous studies demonstrating enhanced forest productivity with N addition (Hyvönen et al., 2008). The tree biomass reduction from 2014 to 2020 was due to thinning in 2015 and affected the organic inputs to soil. Thinning corresponded to a litter input peak, with N+ plots showing higher litterfall than CTR. This difference was confirmed by the above ground litterfall measurements during 2021-2023 (25.1 g $m^{-2}$ $month^{-1}$ in N+ vs. 14.3 g $m^{-2}$ $month^{-1}$ in CTR) (Fig. 2b). Consistently with biomass and litterfall, soil organic carbon (SOC) increased under N fertilization, reaching 5.6 kg C $m^{-2}$ in N+ compared to 4.9 kg C $m^{-2}$ in CTR by 2023 (Fig. 2c), indicating enhanced SOC retention due to reduced microbial respiration (Janssens et al., 2010) alongside aboveground carbon storage in the fertilized stands.

Differences in carbon stocks between treatments prevented drawing conclusions on soil organic matter decomposition rates solely based on the observed increase in heterotrophic respiration ($R_h$) under N fertilization (Fig. 3), and required

normalizing respiration by SOC ($R_{h/SOC}$). When considering normalized respiration, we found a reduction in the annual $R_{h/SOC}$ ratio, rather than an absolute decrease in $R_h$, with fertilization (Fig. 5b). Normalizing respiration by SOC provides a meaningful way to interpret respiration rates relative to carbon availability, especially when comparing treatments with differing SOC stocks. Although this normalization does not fully resolve the issue of $R_h$ dependence on the amount of SOC, it has been widely adopted in field and incubation studies, as well as in soil carbon models (e.g., Tuomi et al., 2008;

Davidson et al., 2012; Curiel Yuste et al., 2007; García-Angulo et al., 2020). Nonetheless, $R_{h/SOC}$ should be interpreted with caution, as it does not capture underlying microbial mechanisms such as enzyme kinetics or community structure, which introduce nonlinearities in the decomposition kinetics. Whereas normalizing $R_h$ by SOC as a proxy of the decomposition rate constant assumes a linear relation between decomposition rate and SOC. In the absence of microbial process data, it serves as a useful, though imperfect, indicator of decomposition rates.

Although, the daily measured mean $R_{h/SOC}$ values were not statistically different between CTR and N+, the $R_{h/SOC}$ responded to N fertilization with reduced sensitivity to soil moisture (Fig. 4), suggesting a potential mechanism (e.g., substrate shifts) for enhanced carbon retention in fertilized plots. However, the slightly increased sensitivity of microbial respiration to temperature at higher values in N fertilized plots (Fig. 4a) may indicate a risk of accelerated carbon loss under warming

conditions in the fertilized soils compared to controls. This dual response to long-term N fertilization, which are discussed in detail in following, highlights the need to consider both moisture and temperature responses in models predicting boreal forest soil carbon dynamics in the context of long-term atmospheric N deposition, fertilization, and climate change.

## 4.1 Response of soil heterotrophic respiration to N fertilization

The meta-analysis of $R_h$ responses to N fertilization in temperate and boreal forests, reported a 15% average decrease in heterotrophic $CO_2$ emissions (Janssens et al., 2010). However, the wide range of responses of heterotrophic $CO_2$ emissions following N fertilization, spanning from a 57% decrease to a 63% increase, encompasses 26% increase in mean soil heterotrophic respiration ($R_h$) from 2021–2023, observed here (Fig. 3a). Limiting $R_h$ by N fertilization in low-productivity forests (Janssens et al., 2010) may relate to low litter quality, as observed in our study's *Calluna*- and *Vaccinium vitis idaea*-

type Scots pine forest.

Yet, higher litter amount due to higher biomass production and thinning in fertilized (N+) than in control (CTR) plots (Fig. 2a and Fig. 2b) may support increased $R_h$ in N+. Although thinning effects on boreal Scots pine $R_h$ are generally modest (Aun et al., 2021), larger inputs of higher-quality litter from harvest residues in N+ plots, including fine roots, needles, and

branches, likely stimulated $R_h$ (Liski et al., 2006; Zhang et al., 2018). This enhanced carbon availability, along with

increased soil nitrogen concentrations, and stimulated microbial activity and biopolymers degradation capabilities explains the observed increase in $R_h$ under N fertilization (Fig. 3). Additionally, we observed a decline in phosphorus concentrations in N-fertilized plots compared to unfertilized plots, probably due to microorganisms mining for phosphorus to sustain their increased activity (Richy et al., 2024).

Despite the significant increase in daily measured $R_h$, the daily measured SOC-normalized heterotrophic respiration ($R_{h/SOC}$) did not differ significantly between CTR and N+ plots but at the annual scale $R_{h/SOC}$ differed. This suggests that increased $R_h$ with N addition originated more from higher litter input and SOC rather than an enhanced microbial decomposition rates (Fig. 3b). The $R_h$ responses to N in Sweden's Rosinedalsheden Scots pine forest also showed variability, with differing
results based on plot size and SOC pool similarity (Hasselquist et al., 2012; Marshall et al., 2021). Using hourly $R_{h/SOC}$ (Curiel Yuste et al., 2007; Shahbaz et al., 2022) may better capture decomposition rate differences than $R_h$ alone, yet hourly-scale $R_{h/SOC}$ responses to N fertilization may still be obscured by fine-scale spatial and temporal variations in soil temperature and moisture (Fig. 3, Fig. S2), primary drivers of $R_{h/SOC}$ (Curiel Yuste et al., 2007; Shahbaz et al., 2022). For example, our biweekly measurements of $R_{h/SOC}$ showed similar means for CTR and N+ plots, but annual $R_{h/SOC}$ means differed (Fig. 6),
reflecting differences in temperature and moisture distribution and differences in functional $R_{h/SOC}$ dependencies to temperature and moisture between treatments (Fig. S1 and Fig. S2; Fig. 4).

## 4.2 Shifts in $R_h$ dependency on soil environmental conditions with N addition

Earth system models often relate $R_h$ to T and SWC, but commonly ignore how the soil N status could modulate such T and
SWC responses (Falloon et al., 2011; Sierra et al., 2015). Here, we observed that N fertilization modified the $R_{h/SOC}$ dependency on both T and SWC, with a sharper increase in $R_{h/SOC}$ with temperature in N+ plots relative to CTR plots. Unlike CTR plots, where $R_{h/SOC}$ declined at temperatures above 15 °C, N+ plots maintained elevated $R_{h/SOC}$ values under high temperatures (Fig. 4a) which is in line with Chen et al. (2024) and may indicate higher risk of increased $CO_2$ emissions from accumulated SOC in warming climates. This increased $R_{h/SOC}$ at high temperature in response to N addition could be
attributed to shifts in substrate composition, where N fertilization enhances the decomposition of labile, C-rich litter (high N availability increases C demand by microbes) and suppresses the decomposition of N-rich organic matter with high lignin content (due to decreased N demand) (Berg and Matzner, 1997; Bonner et al., 2019; Janssens et al., 2010; Wu et al., 2023). Furthermore, our study site exhibited increased Mn peroxidase activity following long-term N addition, indicating enhanced microbial degradation of polyphenolic compounds (Richy et al., 2024). Thus, shifts in litter quality, specifically C and N
ratios, likely contribute to divergent $R_h$ responses to temperature (Robinson et al., 2020).
Moisture also plays a pivotal role in $R_h$ sensitivity to temperature (Robinson et al., 2020), and in modifying soil respiration rates especially under N fertilization and drought conditions (Nair et al., 2024). In our N-fertilized plots, $R_{h/SOC}$ was largely independent of soil moisture, and contrasted with the expected humped response of $R_{h/SOC}$ to SWC in CTR plots (Fig. 4b).

This variation in SWC response suggests potential microbial adaptation to moisture availability (Lennon et al., 2012) and changes in soil physical properties influencing $O_2$ and solute diffusivity (Huang et al., 2023; Moyano et al., 2013). However, the lack of significance of the moisture shape parameter $c$ in the N-fertilized treatment (Table 1) reflected both variability in the data and the absence of a distinct moisture optimum. The contrast between the flat response to moisture in N+ and clear peaked moisture response in CTR highlights potential treatment-related shifts in environmental sensitivity, but also underscores the need to interpret model-based extrapolations with caution. The observed differences between CTR and N+ plots could imply that N status or fertilization-induced changes in soil properties influence the sensitivity of organic matter decomposition to moisture. Soil moisture influences microbial carbon use efficiency (CUE) by affecting substrate accessibility and physiological stress, with lower CUE observed in dry soils (Butcher et al., 2020; Ullah et al., 2021). However, in our study differences in CUE could not be directly inferred from our data, as microbial process measurements were not conducted. Additionally, accelerated decomposition following soil rewetting, commonly referred to as the "Birch effect," has been linked to increased short term N availability (Jarvis et al., 2007). However, prolonged N addition may impose a phosphorus limitation on decomposition (Richy et al., 2024).

Simulating $R_{h/SOC}$ based on both temperature and moisture inputs showed that models relying solely on temperature underestimate $R_{h/SOC}$ for initially N-limited boreal forest soils (Fig. 5). Thus, current soil C models could integrate both temperature and moisture dependencies in their environmental modifiers of decomposition rates, as well as consider variations in SWC response under differing N statuses to improve SOC accuracy in fertile soils (Tupek et al., 2016). For example, the CENTURY model (Parton et al., 1987), which considers topsoil N content and its effect on the fine-litter C ratio, predicts a slight increase in simulated SOC stocks (Tupek et al., 2016), whereas other models like Yasso model (Tuomi et al., 2011) do not account for soil N. However, by restricting topsoil N effects solely to CUE or decomposition rates (Zhang et al., 2018), current models lack the ability to capture the influence of N-driven variations in temperature and moisture modifiers. This limitation highlights the need to re-evaluate the scaling of decomposition with N to better account for the differential respiration sensitivities observed in this study (Fig. 4). Incorporating nonlinear nitrogen effects on temperature and soil moisture modifiers depends on the model's structure. In soil carbon-only models like Yasso, updating these modifiers with a larger dataset that includes nitrogen deposition gradients and soil organic carbon stocks could improve performance. Conversely, in soil carbon-nitrogen models that already account for SOC-N interactions, existing functional relationships should be re-evaluated, considering their interactions with environmental modifiers.

### 4.3 Implications for climate change mitigation

Annually, N-fertilized plots respired 10.3% of SOC (± 0.3 SE), compared to 12.2% (± 0.5 SE) in CTR plots. Although the difference was derived from the modeled values, the lower respiration rate in fertilized plots suggests a potential increase in microbial CUE, which may contribute to long-term SOC accumulation. Despite the winter fluxes not being directly measured, modeled values under low soil temperatures (<5 °C) were close to zero for both treatments due to strong temperature limitation observed in measured data. As a result, differences in winter $R_{h/SOC}$ contributed minimally to annual

estimates and are unlikely to have significantly biased treatment comparisons. This 1.90 (±0.41 SE) % reduction in annual SOC loss due to N addition corresponds to an average of 345.4 (±73.6 SE) g $CO_2$ $m^{-2}$ $y^{-1}$. The combined effect of reduced methane ($CH_4$) oxidation and a slight shift in nitrous oxide ($N_2O$) from a sink to an emitter comparable to Maljanen et al.,

(2006), and equivalent to 8.7 g $CO_2$eq. $m^{-2}$ $year^{-1}$ did not negate this positive mitigation potential and agreed with Öquist et al., (2024). The Haber-Bosch process required for $N_2$ to $NH_3$ conversion has an associated emission cost of approximately 2.96 kg $CO_2$eq. per kg $NH_3$ (Osorio-Tejada et al., 2022), which would reduce our calculated mitigation potential by about 6.5 g $CO_2$ $m^{-2}$ $year^{-1}$. Consequently, the average mitigation potential for N fertilization in our forest soil study is estimated at -327.6 ±73.6 SE g $CO_2$ $m^{-2}$ $yr^{-1}$ (equivalent to 0.89 ±0.2 SE t C $ha^{-1}$ $year^{-1}$). The estimated net GHG mitigation of -327.6 g

$CO_2$ $m^{-2}$ $yr^{-1}$ based on $R_{h/SOC}$ model outputs from a three-year period provides a first-order approximation. However, this estimate does not fully capture the climate impact of fertilization, , as it does not account for longer-term dynamics or potential offsite C and N losses, such as leaching, indirect emissions, or biodiversity-related feedback. Therefore, broader system-level assessments over longer time scales are needed to confirm these findings. While these findings apply to a nutrient-poor boreal ecosystem, extrapolation to similar stands with similar climate—and even more to other ecosystems—

should be done with caution. For example, Saarsalmi et al. (2014) showed that N fertilization stimulated mean annual production (more in nutrient poor pine stands and less in spruce stands with higher nutrient status). Schulte-Uebbing et al. (2021) demonstrated that N addition enhance biomass carbon sequestration primarily in boreal regions, while having negative effects in tropical forests.

**5 Conclusions**

Although our experiment design allowed exploratory insights into N fertilization effects, caution is needed in extrapolating beyond this site. While results represent a case study, they reveal that increased soil N status after long-term N fertilization in boreal Scots pine ecosystems can alter the dependency of C decomposition on temperature and moisture. The results also suggest a net reduction in soil GHG emissions with long-term N fertilization, indicating that N fertilization in our boreal

Scots pine forest not only enhanced tree biomass but may also acted as a viable forest management strategy for climate change mitigation.

**Acknowledgements**

The study was conducted in the HoliSoils project (Holistic management practices, modelling and monitoring for European forest soils) funded by the European Union's Horizon 2020 research and innovation program (Grant Agreement No.

101000289). This study has been done with affiliation to the UNITE Flagship funded by the Research Council of Finland (decision 357909). We thank our field team lead by Petri Salovaara for collecting high-quality measurements. We also thank

Mikko Kukkola and Hannu Ilvesniemi for the tree biomass monitoring data. We used BioRender for designing the graphical abstract. We appreciate constructive comments of Marleen Pallandt and the referees.

**Data and code availability**

Complete data set on GHG fluxes, soil temperature and moisture, tree biomass and litter production, and soil carbon stocks are archived and available on ZENODO (https://doi.org/10.5281/zenodo.13889762). The R scripts supporting results replication is also openly available on ZENODO (https://doi.org/10.5281/zenodo.14101488).

**Author contribution**

BT, AL, RM designed the hypothesis and experimental design. RM and AL arranged research funding and oversaw project
management. BT contributed to data collection and carried out the analysis. BT prepared the manuscript with contributions from all co-authors.

**Competing interests**

Some authors are members of the editorial board of journal Biogeosciences.

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
