# Peer review of "Long-term nitrogen fertilization alters microbial respiration sensitivity to temperature and moisture, potentially enhancing soil carbon retention in a boreal Scots pine forest"

_EGUsphere, 2024_

## Author Comment (AC1)

**Preprint egusphere-2024-3813 Discussion:** Reduced microbial respiration sensitivity to soil moisture following long-term N fertilization enhances soil C retention in a boreal Scots pine forest

**Ťupek et al. reply to RC1**: 'Comment on egusphere-2024-3813', Anonymous Referee #1, 02 Jan 2025

Corresponding author: [boris.tupek@luke.fi](mailto:boris.tupek@luke.fi)

*Authors' responses are in Italics (and in dark-blue color in the attached pdf version, which also includes revised figures and tables). Revised text is indicated by quotation marks.*

The manuscript raises important questions about the effects of N fertilization on soil GHG emissions and moisture dynamics. The authors have collected unique and valuable data. The paper is mainly very clear and well written. I have read the manuscript with great interest, but I am concerned about several issues in the methodology, results, and conclusions. The data presented do not adequately support the conclusions, and there are inconsistencies between the observations and the model outputs. Below, I outline the main issues with the study followed by few detailed comments.

*Thank you for your thorough review and constructive comments, which have helped improve our manuscript. Below, we provide detailed responses to each remark, along with corresponding revisions.*

**Conflicting patterns in observations**

In Supplementary Figure S2, the highest mean momentary soil moisture values appear to be in the N+ treatment during 2021–2022. However, in Figures 3F and 4B, the highest soil moisture values are in the CTR treatment. While these observations may not be strictly contradictory, they are at least unusual and require further explanation. Additionally, in 2022, the soil moisture appears lowest in N+ (S2), while in 2021 and 2023, it is approximately equal with CTR. Despite this, Figure 4B suggests there are notably more low soil moisture values (<0.2) in CTR than in N+. These discrepancies between the observations and figures need clarification.

*Discrepancies (especially in extreme values) could be explained by different nature of data points. Whereas in Figure S2 data represent weekly averages at the treatment level in Figures 3F and 4B the data points are the instantaneous measurements of individual plots of the treatments.*

It also seems very strange that the soil moisture has such a different dynamic in the different treatments in late July 2023, with N+ zigzagging (Fig S2). I recommend checking the dates for N+.

*We checked the data in Fig. S2, and the date of measurements is correct.*

*To improve data interpretation, we revised Fig. 1 by adding a new panel (B) that illustrates topographic variation and distances between treatment plots.*

[Figure]

**Figure 1:** *Geographical location of the Karstula forest study site in Finland (A); topographical variation of the study site and the location of treatment (control CTR and N-fertilized N+) plots (B), photograph of the forest stand (C); and one of six 2 x 1 m forest floor plot groups, each with four subplots used for measuring soil greenhouse gases, soil organic C, and soil temperature and moisture following the installation of a root-exclusion fabric (D).*

*We also revised text in results by including following sentence:*

*" High variation in soil moisture between CTR and N+ plots (located on average 122 m apart) could be attributed to the measured topsoil humus layer being highly affected by microscale variations of vertical and lateral water flows due to variable microtopography and tree canopy openings (Fig. 1B)."*

**Conflicting results by the observations and the model**

The authors themselves state based on the observations that 'Rh showed sensitivity to T and SWC, rising with warmer conditions and declining in dry periods, then recovering after rewetting events (Fig. S2).'

*To improve the clarity in temporal patterns between treatments the text in results was revised:*

*"$R_h$ showed sensitivity to T and SWC, generally rising with warmer conditions and declining in dry periods, then recovering after rewetting events (Fig. S2). However, this pattern was more pronounced in CTR than in N+ plots."*

Then, in the results and discussion they state that 'Soil moisture effects on Rh/SOC were only observed in control (CTR) plots' and 'In our N-fertilized plots, Rh/SOC was largely independent of soil moisture.'

*With the added sentence above, this now aligns with the message in the results.*

These latter ones arise from the model that is fitted to the growing season (Apr-Oct) data and then applied for the whole year. These estimates are presented without any estimates of uncertainty, which undermines its reliability.

*To enhance model reliability, we replaced the stepwise fitting approach with a single model where both temperature and moisture vary simultaneously (see details below).*

*In the revised text, we have included standard errors (SE) in the presented values in the abstract, updated Chapter 3.6 with a new table of annual GHG and GWP estimates, and revised Chapter 4.3.*

*Additionally, we provide SE and p-values for estimated parameters in Table 1, goodness-of-fit statistics (R², RMSE, MBE, MAE) in Table 2, and standard deviations (SD) for cross-validation goodness-of-fit statistics (SD of R², RMSE, and MAE) in Table S1. Error bars (SE) are also included in Figure 6.*

I see that the main issue behind this is that the models presented in the study do not perform well, especially during the dry year of 2021 (Figure 5). In the figure, for example, the observed soil moisture data for N+ in 2021 show a significant mid-season drop, which is as substantial, if not even larger than the drop in CTR. However, the model fails to capture this, and based on the model, the authors conclude that there is no soil moisture effect. In other words, the main conclusion of the whole paper appears to be unsupported by the observational data and raises questions about the validity of the model. The poor performance might arise from the model structure but also that there are significantly fewer data points for the N+ treatment under very wet conditions (>0.45 SWC, just two observations). Similarly, under dry conditions (<0.2), there are notably more measurements for the CTR treatment than for the N+ treatment. This limited dataset could easily skew the soil moisture response curve.

*Given your remark, to have more confidence in the model, we revised our stepwise modeling approach—where temperature was fitted first, followed by temperature and moisture—which may have impacted the observed temperature optimum in control and hindered relations to moisture in N fertilized plots. In the revised model, we fitted the full temperature and moisture model in one step where we have allowed both temperature and moisture vary simultaneously. Thus, in the revised paper we evaluate only temperature-moisture model, as both variables are correlated and impact microbial respiration simultaneously.*

*The text describing the model was revised as well as all results. "NLS regression was used to extrapolate $R_{h/SOC}$ to continuous hourly data and to upscale $R_{h/SOC}$ to the annual level. The combined T and SWC dependency of $R_{h/SOC}$ was modeled by multiplying a Gaussian T function as described in Tuomi et al. (2008) with a Ricker function for SWC (Bolker, 2008) (Eq. 1):*

$$R_{h/SOC}(T, SWC) = e^{(\beta_1 T + \beta_2 T^2)}(a\, SWC\, e^{(-b\, SWC)})^c \, , \qquad\qquad (1)$$

*where β₁ and β₂ are parameters controlling the exponential T response, and parameters a determine the initial slope, b the post-optimal decline, and c the peak height of SWC response."*

*However, the results of the revised model (the shape of the temperature and moisture responses) are close to those from the preprint (see updated Fig. 4 below).*

[Figure]

***Figure 4: (A) Dependence of soil microbial respiration normalized by soil organic carbon (R_h/SOC, μg C g⁻¹ SOC h⁻¹) on soil temperature at 5 cm depth (T, °C). (B) Ratio of measured R_h/SOC to modeled R_h/SOC(T,SWC_mean) as a function of volumetric water content (SWC, m³ m⁻³) at 5 cm depth. Panels display results separately for control (CTR) and N-fertilized (N+) plots. Shading of turquoise points in (A) reflects varying SWC, while shading of red points in (B) corresponds to variation in T. Black lines indicate local polynomial regression (LOESS) fits with gray ribbons showing 95% confidence intervals; yellow lines represent nonlinear least square (NLS) regression model fits.***

*The model performance for the N+ plots simulations improved for year 2021, being able to better reconstruct reduction of respiration during the summer drought event (see updated Fig. 5 below).*

[Figure]

***Figure 5: Time series of daily mean R_h/SOC (μg C g⁻¹ SOC h⁻¹) in CTR and N+ plots, with measurements shown as points (error bars indicate SE) and model estimates as lines (ribbons indicate SE).***

*To improve the interpretation of results we revised the text in discussion accordingly:*

*"Due to differences in carbon stocks between treatments, decomposition rates expressed as $R_h$ are not directly comparable between CTR and N+. Therefore, drawing conclusions on respiration rate differences required normalizing respiration by SOC. However, high variability in momentary $R_{h/SOC}$ measurements prevented definitive conclusions, as the mean $R_{h/SOC}$ values were not statistically different between CTR and N+.*

*In contrast, model parameters describing functional dependencies on soil moisture were statistically significant for CTR but not for N+. However, neither the CTR nor N+ models showed bias (Table 2 and S1). Differences in functional forms between CTR and N+ (Fig. 3) led to lower annual respiration estimates for N+ compared to CTR (Fig. 6)."*

*Based on the model results, we conclude that respiration sensitivity to soil moisture was reduced in N+ compared to CTR.*

*To address your remark that respiration in N+ was also reduced during severe drought, we revised the title to: 'Long-term nitrogen fertilization alters microbial respiration sensitivity to temperature and moisture, potentially enhancing soil carbon retention in a boreal Scots pine forest.'*

**Misleading conclusions about GHG emissions**

The authors conclude: 'Our results also suggest a net reduction in soil GHG emissions with long-term N fertilization.'

However, this is only supported by the poor model and not by the observational data, which show increased Rh (with no difference in Rh/SOC), reduced CH4 sink and increased N2O emissions in the fertilized treatment compared to the control. These observations indicate that the net impact of N fertilization on soil GHG emissions may be neutral or even negative. To see the relative importance of CH4 and N2O, it would be useful for the reader to see all fluxes as CO2 equivalents.

*The inability to compare Rh means directly (due to SOC differences between treatments) as well as the need for model in annual upscaling of Rh/SOC (due to T, SWC seasonality) was explained previously. However, to increase confidence in Rh/SOC model results in the revised paper we improved the model fitting to allow temperature and moisture vary simultaneously resulting in better match between the model estimates and measurements (Fig. 5). We also clarified in conclusions reliability of the model by adding following sentence:*

*"Although the models showed relatively large mean residuals when evaluated against individual measurements, their mean bias errors were close to zero (Table 2)."*

*In the preprint, we mention the contribution of different processes to $CO_2$ emissions reduction potential after long-term N fertilization in abstract L27-29 and in discussion L349-353. In the revised text, we also help the reader to visualize the relative contributions of individual processes in $CO_2$-equivalents as GWP-100 potentials by detailing these results into a new table (Table 3) in revised chapter 3.6.*

**Table 3: Annual global warming potential (GWP) reduction by long-term N fertilization in boreal Scots pine forest with contribution of individual greenhouse gas (GHG) fluxes (microbial respiration normalized by soil organic carbon stock $R_{h/SOC}$, $CH_4$ net oxidation, and $N_2O$ flux net exchange) evaluated as a difference between control (CTR) and N fertilized plots (N+). Minus values indicate net GWP reduction. The GWP-100 values (27 for $CH_4$ and 273 for $N_2O$) were used for calculation of $CO_2$-equivalents.**

| Treatment | $R_{h/SOC}$ (%) | | GWP-$CO_2$ (g $CO_2$ m$^{-2}$ y$^{-1}$) | | $CH_4$ (g $CH_4$ m$^{-2}$ y$^{-1}$) | | GWP-$CH_4$ (g $CO_2$-eq m$^{-2}$ y$^{-1}$) | | $N_2O$ (mg $N_2O$ m$^{-2}$ y$^{-1}$) | | GWP-$N_2O$ (g $CO_2$-eq m$^{-2}$ y$^{-1}$) | | GWP-GHG (g $CO_2$-eq m$^{-2}$ y$^{-1}$) | |
|---|---|---|---|---|---|---|---|---|---|---|---|---|---|---|
| | mean | SE | mean | SE | mean | SE | mean | SE | mean | SE | mean | SE | mean | SE |
| **CTR** | 12.2 | 0.5 | 2214.9 | 90.4 | -1.6 | 0.0 | -42.9 | 0.5 | -2.2 | 0.8 | -0.6 | 0.2 | 2171.4 | 90.4 |
| **N+** | 10.3 | 0.3 | 1869.5 | 56.8 | -1.2 | 0.0 | -32.8 | 0.5 | 1.9 | 0.6 | 0.5 | 0.2 | 1837.3 | 56.8 |
| **Difference** | -1.9 | 0.4 | **-345.4** | **73.6** | 0.4 | 0.0 | **10.1** | **0.5** | 4.2 | 0.7 | **1.1** | **0.2** | **-334.1** | **73.6** |

**Issues with Methods**

The methods section contains critical gaps that limit the reproducibility and reliability of the study:

The manuscript does not describe the equations or models used for tree biomass in the methods but refer to Lehtonen et al. in the Fig 2 caption. The methods section should be improved here.

*The Fig2a caption was revied "(A) Estimated tree biomass and litterfall from 1980 to 2020 forest tree stands inventory measurements."*

*The biomass equations and litter turnover rates models citing Repola (2009) and Lehtonen et al. (2016) were detailed in chapter "2.2.1 Tree inventory and litterfall".*

The manuscript does not include any estimates of uncertainty for their main result that is the decreased (modelled) Rh/SOC after fertilization.

*In revised paper we added standard error (SE) in presented values of abstract, and updated Chapters 3.6 and 4.3. The SE of main results are also provided in Fig. 6.*

Fluxes were measured over only 3 minutes using a large (21 L) chamber. Considering the low flux rates of CH4 and N2O, it is doubtful whether this short measurement duration is sufficient for reliable estimates. It might be though but makes me worry if the equipment used is sensitive enough to detect such low fluxes. Of course, it is not the volume that is important here, but the area, but this is not given. Based on the chamber description in the manuscript, no one could repeat it.

Other

*The diameter of measurement plots (30 cm) thus of the chamber is mentioned in the preprint on line 112. We revised the text by including dimension of 30 cm diameter in chamber description.*

*To improve the confidence on the precision of the GHG measurements, we revised the paper by adding description of measurement method detection limits:*

*"The $CH_4$ and $N_2O$ concentrations were measured during 3 min intervals with 5 second averaging at the 0.25 ppb precision for $CH_4$ and 0.20 ppb precision for $N_2O$. The minimum detectable flux of measurements estimated using the formula by Parkin et al., (2012) was 0.0238 $\mu g\ m^{-2}\ h^{-1}$ for $CH_4$ and 0.0524 $\mu g\ m^{-2}\ h^{-1}$ for $N_2O$.*

*Parkin, T.B., Venterea, R.T., Hargreaves, S.K., 2012. Calculating the detection limits of chamber-based soil greenhouse gas flux measurements. J. Environ. Qual. 41, 705–715."* https://doi.org/10.2134/jeq2011.0394

*We clarified in results that:*

*"The method detection limits were smaller than SE of mean $CH_4$ and $N_2O$ fluxes."*

The authors state in the introduction: 'Moreover, full accounting of GHG emissions should include emissions associated with N fertilizer production.' However, they do not include these emissions in their own analysis and conclusions.

*The associated emissions with fertilizers productions were accounted for according to Osorio-Tejada et al. (2022) and presented in chapter 4.3 lines 351-353.*

*We revised methos by adding:*

*"The associated emissions with fertilizers productions were accounted for according to Osorio-Tejada et al. (2022). We estimated the $CO_2$ emissions associated with six nitrogen fertilization events, which occurred once per decade between 1960 and 2020. The applied nitrogen fertilization rate was 180 kg N ha $^{-1}$ per event. Converting this to ammonia ($NH_3$) using the molecular weight ratio of $NH_3$ to N (17.031/14.007) resulted in an estimated 218.86 kg $NH_3$ ha $^{-1}$ per fertilization event. Given an emission factor of 2.96 kg $CO_2$ per kg $NH_3$, this corresponds to 647.93 kg $CO_2$ ha $^{-1}$ per event. Over six fertilization events spanning 60 years,*

*the annualized $CO_2$ emission was calculated as 64.79 kg $CO_2$ ha$^{-1}$ yr$^{-1}$, equivalent to approximately 6.5 g $CO_2$ m$^{-2}$ yr$^{-1}$."*

*Osorio-Tejada, J., Tran, N.N., Hessel, V., 2022. Techno-environmental assessment of small-scale Haber-Bosch and plasma-assisted ammonia supply chains. Science of The Total Environment 826, 154162. https://doi.org/10.1016/j.scitotenv.2022.154162*

Although CH4 and N2O fluxes are a central part of one of the hypotheses, those are not well motivated and the discussion does not address these at all.

*We revised the hypothesizes "We hypothesized that (i) increased soil nitrogen availability would enhance soil organic carbon (SOC) accumulation and heterotrophic respiration ($R_h$) due to greater biomass growth and litter inputs, while SOC-normalized $R_h$ ($R_{h/SOC}$) would decline due to reduced microbial nitrogen demand; and (ii) nitrogen fertilization would alter $CH_4$ oxidation and increase $N_2O$ emissions compared to N-limited soils, reflecting shifts in microbial activity and substrate availability."*

*Reasoning for the hypothesis (ii) is detailed on lines 46-49. We discussed these findings shortly in Chapter 4.3 lines 349 -351 in relations to studies by Maljanen et al., (2006), and Öquist et al., (2024).*

*We revised text in chapter 4.3 by adding: "Although, the $CH_4$ and $N_2O$ fluxes need consideration due larger GWP than CO2 and potentially large N2O fluxes after fertilization, the CH4 and N2O fluxes observed in our study were very close to zero thus showed negligible contribution to total forest soil GHG emissions."*

The title and main findings revolve around soil moisture dependency, yet this was not one of the original hypotheses or a focus in the introduction. This shift in focus feels post-hoc, as if it were added after analyzing the data and models, rather than being a central research question from the start. For that reason, the story does not seem to hold together.

*The title in preprint reflects the main findings supporting hypothesis (i) expected changes in respiration and soil C stock after fertilization.*

*Considering your remark that during severe drought period observed respiration of N+ was also reduced, we revised the tittle: "Long-term nitrogen fertilization alters microbial respiration sensitivity to temperature and moisture, potentially enhancing soil carbon retention in a boreal Scots pine forest".*

Detailed comments, by line number

15: Carbon (C) is usually written out in full the first time it is mentioned, as was done for nitrogen (N) even though we all know it.

*We changed C to carbon.*

30-32: The conclusion seems overly broad, given that just one upland forest was studied. Your site was originally a very poor Scots pine forest, but you generalize your conclusions to all forest types. What about peatland forests? Do you know, even for your own site, what, for example, the N2O fluxes were just after the fertilisation events or in earlier phases of the rotation?

*To narrow down the concluding sentence of the abstract we replaced "boreal forest" by "boreal Scots pine forests on mineral soils" and in revised text of conclusions chapter we replaced "boreal Scots pine ecosystems" by "boreal Scots pine forests on mineral soils".*

73-74: These seem like very nice references, but I'm not sure that they are both conducted in the actual boreal zone and represent the entire boreal zone?

*Yes, these studies were conducted in Southern Finland and Estonia, within the boreal and hemi-boreal zones, representing conditions of the southern boreal region. The sentence was revised. "In southern boreal region's Scots pine forests on well-drained mineral soils, ..."*

L81: The introduction lacks any reasoning/motivation for such a hypothesis.

*Reasoning for the hypothesis is detailed on lines 46-49.*

L89: Even if you follow the silvicultural practices in principle, there can be a lot of variation in practice. So for the sake of repeatability I would add some details on the harvests, like how much basal area was reduced or something like that.

*The sentence was revised to include information on basal area:*

*"The stand underwent thinning in 1990 (reducing 16.2% and 26.5% of basal area (BA) for CTR and N+, respectively), and 2015 (reducing 36.7 % and 40.1% of BA for CTR and N+, respectively)."*

111: Please be more specific and use dates instead of growing seasons.

*In revised text we replaced the growing seasons by exact dates.*

112: I don't understand. Did you take measurements from two individual points within each of three or six plots, which you refer to as a group? If you had two points, is that a group or a pair? How close were the groups to each other? Are they independent? How do you take into account in the statistical analyses that the two points are close and probably not independent? Please clarify the description of the overall setup including what was the distance between the points and groups and treatments.

*In revised text we reformulated "12 plot groups (two 30 cm diameter plots per group; n=6 per treatment)." to "12 plots (6 plots or 3 pairs per treatment). Plots in each pair were located 30 cm apart (Fig. 1c) and CTR and N pairs were on average 122 m apart (Fig. 1b)."*

*Location of the plots resulted from the experimental setup of the fertilization treatments. In the revised text we clarified:*

*"As the single plot area was relatively large (706 cm2), we considered 2 plots pair to be representative of the trenched area (Fig. 1c) and 3 pairs to be representative of the spatial variation of the treatment."*

114: Why do you use both Rh and $R_h$ for heterotrophic respiration here and elsewhere? It gives a slightly unfinished impression.

*$R_h$ is correct and in revised text we corrected Rh in all instances.*

116-119: This paragraph seems to be the earlier or at least less complete version of the following one, please combine these sections to avoid redundancy.

*In revised text the lines 116 – 119 were combined with lines 121-129.*

116&122: Don't you need to write down the manufacturer's details anymore?

*In revised text we added manufacturer's details "(LICOR, Lincoln, NE, USA)".*

131 Depth should be given

*L131 mentions depth "at 5 cm depth"*

132 end date is missing

*In revised text the end date was added "until end of December 2023"*

142 You have not yet introduced CTR and N+

*in revised text the abbreviations were explained at the first instance*

---

## Author Comment (AC2)

**Preprint egusphere-2024-3813 Discussion:** Reduced microbial respiration sensitivity to soil moisture following long-term N fertilization enhances soil C retention in a boreal Scots pine forest

**Ťupek et al. reply to RC2:** 'Comment on egusphere-2024-3813', **Anonymous Referee #2, 25 Feb 2025**

Corresponding author: [boris.tupek@luke.fi](mailto:boris.tupek@luke.fi)

*Authors' responses are in Italics (and in dark-blue color in the attached pdf version, which also includes revised figures and tables). Revised text is indicated by quotation marks.*

This study investigates the effects of long-term nitrogen (N) fertilization on soil heterotrophic respiration (Rh), methane ($CH_4$) oxidation, and nitrous oxide ($N_2O$) emissions in a boreal Scots pine forest. The results show that N fertilization increased tree biomass, litterfall, and soil organic carbon (SOC) stocks. Despite elevated Rh in fertilized plots, SOC-normalized Rh (Rh:SOC ) did not differ significantly between fertilized (N+) and control (CTR) plots. N fertilization altered Rh:SOC dependencies on temperature (T) and soil water content (SWC), with N+ plots exhibiting increased temperature sensitivity and reduced SWC dependence. These shifts, combined with reduced $CH_4$ oxidation and increased $N_2O$ emissions, resulted in a net reduction in soil greenhouse gas (GHG) emissions, suggesting enhanced SOC retention under N fertilization. The findings highlight the potential of long-term N fertilization to mitigate climate warming in boreal forests by altering microbial respiration dynamics. There are some interesting findings in this study, but I have several major comments and suggestions based on the current version of manuscript. Please find them below.

*Thank you for your thorough review and constructive comments, which have helped improve our manuscript. Below, we provide detailed responses to each remark, along with corresponding revisions.*

The study reports increased Rh under N fertilization, contrasting with the widely observed suppression of soil $CO_2$ respiration by N addition. While this is an innovative discovery, further mechanistic explanations are needed. For example, how do microbial community shifts (e.g., enzyme activities, CUE, substrate quality) under N fertilization drive this response? Is this phenomenon unique to nutrient-poor boreal systems, or could it apply to other ecosystems? A broader discussion of context-dependent mechanisms is warranted.

*The response of microbial activity and substrate quality to N addition has been investigated in Karstula and detailed in Richy et al. (2024). In revised paper, we expanded discussion by highlighting following mechanisms:*

*"Nitrogen addition stimulated tree biomass production, which in turn increased carbon inputs into the soil. This enhanced carbon availability, along with increased soil nitrogen*

*concentrations, stimulated microbial activity and biopolymers degradation capabilities. This process certainly explains the observed increase in heterotrophic respiration (Rh) under N fertilization. Additionally, we observed a decline in phosphorus concentrations in N-fertilized plots compared to unfertilized plots, probably due to microorganisms mining for phosphorus to sustain their increased activity."*

The conclusions are based on a single long-term experimental site. While valuable, this limits generalizability. Are there similar responses observed in other boreal forests? The authors may need to address how site-specific factors (e.g., soil type, microbial composition, thinning history) might influence their results.

*In revised paper, we expanded discussion on generalizability of our results:*

*"While these findings likely apply to nutrient-poor boreal ecosystems, extrapolation should be done for similar stands with similar climate with caution for other ecosystems. For example, Saarsalmi et al. (2014) showed that N fertilization stimulated growth in relation to mean annual production (more in nutrient poor pine stands and less in spruce stands with higher nutrient status). Schulte-Uebbing et al. (2021) demonstrated that N addition enhance biomass carbon sequestration primarily in boreal regions, while having negative effects in tropical forests."*

*Saarsalmi, A., Smolander, A., Moilanen, M., Kukkola, M., 2014. Wood ash in boreal, low-productive pine stands on upland and peatland sites: Long-term effects on stand growth and soil properties. Forest Ecology and Management 327, 86–95. https://doi.org/10.1016/j.foreco.2014.04.031*
*Schulte-Uebbing, L.F., Ros, G.H., de Vries, W., 2022. Experimental evidence shows minor contribution of nitrogen deposition to global forest carbon sequestration. Global Change Biology 28, 899–917. https://doi.org/10.1111/gcb.15960*

This paper mentioned global warming potential (GWP), which can be calculated from fluxes of CO2, CH4, and N2O, but they did not calculate based on their data. It would be interesting to examine how GWP was changed under the long-term N fertilization. While reduced $CH_4$ oxidation and increased $N_2O$ emissions are noted, their combined contribution to global warming potential (GWP) is dismissed as minor. Some discussions could be added for whether long-term N fertilization could eventually offset SOC gains through cumulative $N_2O$ emissions.

*Yes, our study found that in N-fertilized plots, the impact of reduced $CH_4$ oxidation and increased $N_2O$ emissions played a minor role compared to the reduction in $CO_2$ emissions.*

*For improved clarity in the revised version of the paper, we calculated the global warming potential (GWP) using the AR6 GWP-100 values (27 for $CH_4$ and 273 for $N_2O$) and added the corresponding text to the methods and results sections along with a new table (Table 3) in revised chapter 3.6.*

*Intergovernmental Panel On Climate Change (IPCC), 2023. Climate Change 2021 – The Physical Science Basis: Working Group I Contribution to the Sixth Assessment Report of the Intergovernmental Panel on Climate Change, 1st ed. Cambridge University Press. https://doi.org/10.1017/9781009157896*

**Table 3: Annual global warming potential (GWP) reduction by long-term N fertilization in boreal Scots pine forest with contribution of individual greenhouse gas (GHG) fluxes (microbial respiration normalized by soil organic carbon stock $R_{h/SOC}$, $CH_4$ net oxidation, and $N_2O$ flux net exchange) evaluated as a difference between control (CTR) and N fertilized plots (N+). Minus values indicate net GWP reduction. The GWP-100 values (27 for $CH_4$ and 273 for $N_2O$) were used for calculation of $CO_2$-equivalents.**

| Treatment | $R_h/SOC$ (%) | | GWP-$CO_2$ (g $CO_2$ $m^{-2}$ $y^{-1}$) | | $CH_4$ (g $CH_4$ $m^{-2}$ $y^{-1}$) | | GWP-$CH_4$ (g $CO_2$-eq $m^{-2}$ $y^{-1}$) | | $N_2O$ (mg $N_2O$ $m^{-2}$ $y^{-1}$) | | GWP-$N_2O$ (g $CO_2$-eq $m^{-2}$ $y^{-1}$) | | GWP-GHG (g $CO_2$-eq $m^{-2}$ $y^{-1}$) | |
|---|---|---|---|---|---|---|---|---|---|---|---|---|---|---|
| | mean | SE | mean | SE | mean | SE | mean | SE | mean | SE | mean | SE | mean | SE |
| **CTR** | 12.2 | 0.5 | 2214.9 | 90.4 | -1.6 | 0.0 | -42.9 | 0.5 | -2.2 | 0.8 | -0.6 | 0.2 | 2171.4 | 90.4 |
| **N+** | 10.3 | 0.3 | 1869.5 | 56.8 | -1.2 | 0.0 | -32.8 | 0.5 | 1.9 | 0.6 | 0.5 | 0.2 | 1837.3 | 56.8 |
| **Difference** | -1.9 | 0.4 | **-345.4** | **73.6** | 0.4 | 0.0 | **10.1** | **0.5** | 4.2 | 0.7 | **1.1** | **0.2** | **-334.1** | **73.6** |

The increased temperature sensitivity of Rh:SOC in N+ plots (Fig. 4a) raises concerns about accelerated SOC loss under warming. It is interesting that the optimum temperature emerged under control but not under N+ treatment. Can you please explain why this happens? The current version only mentioned this pattern based on modeling results (lines ~210).

*Regarding the emergence of a temperature optimum within the observed soil temperature range, in addition to the mechanisms discussed on lines 314–323, the temperature optimum (with limitations beyond the peak) in the control plots could partly be attributed to moisture limitation at higher temperatures.*

*In the preprint, our stepwise modeling approach—where temperature was fitted first, followed by temperature and moisture—may have amplified this effect. In the revised approach, we have allowed both temperature and moisture to vary simultaneously thus fitted both functions together in one model.*

*The text describing the model was revised as well as all results.*

*"NLS regression was used to extrapolate $R_{h/SOC}$ to continuous hourly data and to upscale $R_{h/SOC}$ to the annual level. The combined T and SWC dependency of $R_{h/SOC}$ was modeled by multiplying a Gaussian T function as described in Tuomi et al. (2008) with a Ricker function for SWC (Bolker, 2008) (Eq. 1):*

$$R_{h/SOC}(T, SWC) = e^{(\beta_1 T + \beta_2 T^2)}(a\,SWC\,e^{(-b\,SWC)})^c , \qquad (1)$$

*where β₁ and β₂ are parameters controlling the exponential T response, and parameters a determine the initial slope, b the post-optimal decline, and c the peak height of SWC response."*

*The results of the revised model (the shape of the temperature and moisture responses) are close to those from the preprint (see updated Fig. 4 and 6 below, and Table 3). However, the temperature optimum with the revised model was observed for both CTR and N+ plots at 15.8 °C and 16.8 °C, respectively.*

[Figure]

*Figure 4: (A) Dependence of soil microbial respiration normalized by soil organic carbon ($R_{h/SOC}$, µg C g⁻¹ SOC h⁻¹) on soil temperature at 5 cm depth (T, °C). (B) Ratio of measured $R_{h/SOC}$ to modeled $R_{h/SOC}(T,SWC_{mean})$ as a function of volumetric water content (SWC, m³ m⁻³) at 5 cm depth. Panels display results separately for control (CTR) and N-fertilized (N+) plots. Shading of turquoise points in (A) reflects varying SWC, while shading of red points in (B) corresponds to variation in T. Black lines indicate local polynomial regression (LOESS) fits with gray ribbons showing 95% confidence intervals; yellow lines represent nonlinear least square (NLS) regression model fits.*

[Figure]

*Figure 6: Annual $R_{h/SOC}$ (% SOC per year) estimated with NLS models driven by combined temperature and moisture ($R_{h/SOC}(T, SWC)$, Eq. 1) in CTR and N+ plots, using hourly T and SWC data for model inputs (Fig. S1).*

Minor comments:

Page 1, Line 15: change "Nutrient availability effects microbial respiration kinetics" to "Nutrient availability affects microbial respiration kinetics."

*changed as suggested*

Page 4, Line 75: Clarify the source of GWP values (23 for $CH_4$, 296 for $N_2O$). Are these IPCC AR5 or AR6 values? Update citation if necessary.

*GWP values (on Page 4, Line 75) were from Ramaswamy et al. (2019)*

*Ramaswamy, V. et al., 2019: Radiative Forcing of Climate: The Historical Evolution of the Radiative Forcing Concept, the Forcing Agents and their Quantification, and Applications. Meteorological Monographs, 59, 14.1–14.101, doi:10.1175/amsmonographs-d-19-0001.1.*

*We revised version we used AR6 GWP-100 values (27 for $CH_4$, 273 for $N_2O$) and updated the citation.*

*Intergovernmental Panel On Climate Change (Ipcc), 2023. Climate Change 2021 – The Physical Science Basis: Working Group I Contribution to the Sixth Assessment Report of the Intergovernmental Panel on Climate Change, 1st ed. Cambridge University Press. https://doi.org/10.1017/9781009157896*

Page 5, Line 95: Provide details on measurement protocols (e.g., instruments used, precision) for tree diameter, height, and crown base height.

*In revised text we updated the tree measurement protocols accordingly:*

*"In each CTR and N+ plot, the breast-height diameter (d1.3) of all trees has been measured using a caliper (±1 mm precision) once per decade since 1960, as well as after the 2015 thinning. Additionally, in a permanent subset of trees representing various size categories, tree height and crown base height have been recorded using a hypsometer (precision ~0.5–1 m)."*

Page 6, Line 125: Elaborate on how chamber headspace linearity was monitored. Was a threshold $R^2$ value used to accept/reject flux calculations?

*We revised text accordingly: "…, and linearity was monitored visually during the measurements to accept only fluxes with increasing trends in $CO_2$ concentration evolution."*

Page 8, Line 185: Report ANOVA statistics (F-value, degrees of freedom, p-value) for Rh differences between N+ and CTR.

*We added ANOVA statistics (F-value, degrees of freedom, p-value):*

*"Pairwise ANOVA showed that mean annual soil microbial $R_h$ (g $CO_2$ m$^{-2}$ h$^{-1}$) was significantly higher in N+ (0.58 ± 0.01 SE) than in CTR plots (0.46 ± 0.01 SE) (F-value 15.96, degrees of freedom 449, p-value 8.92e-05) (Fig. 3a)."*

Page 10, Line 220: Add $R^2$ values or confidence intervals to Table 1/Table 2 to quantify model explanatory power.

*We added $R^2$ values to Table 2. Please note, that with one improved model where both temperature and moisture vary simultaneously, there was no reason for AIC and BIC model comparison statistics.*

***Table 2: Goodness-of-fit statistics for NLS models based combined temperature and SWC (Eq. 1): proportion of explained variance ($R^2$), root mean square error (RMSE), mean bias error (MBE), and mean absolute error (MAE). RMSE, MBE and MAE in µg C g$^{-1}$ SOC h$^{-1}$.***

| Treatment | $R^2$ | RMSE | MBE | MAE |
|-----------|-------|------|-----|-----|
| | | µg C g$^{-1}$ SOC h$^{-1}$ | | |
| CTR | 0.41 | 15.55 | -0.33 | 11.42 |
| N+ | 0.40 | 13.36 | -0.48 | 9.28 |

Page 14, Line 280: Expand the discussion on why N fertilization reduced SWC sensitivity (e.g., microbial adaptation, substrate shifts).

*We reformulated the Line 280 on why N fertilization reduced SWC sensitivity:*

*"In our study, soil microbial respiration responded to N fertilization with a reduced sensitivity to soil moisture (Fig. 4b), suggesting a potential mechanism (e.g., microbial adaptation, substrate shifts) for enhanced carbon retention in fertilized plots which are discussed in detail in following chapters."*

Page 16, Line 340: Suggest specific model improvements (e.g., incorporating nonlinear N effects on T/SWC modifiers) to address current limitations.

*Thank you for request to elaborate this interesting point. We added following sentences into the discussion:*

*"Incorporating nonlinear nitrogen effects on temperature and soil moisture modifiers depends on the model's structure. In soil carbon-only models like Yasso, updating these modifiers with a larger dataset that includes nitrogen deposition gradients and soil organic carbon stocks could improve performance. Conversely, in soil carbon-nitrogen models that already account for SOC-N interactions, existing functional relationships should be re-evaluated, considering their interactions with environmental modifiers.*

---

## Author Response (AR2)

**Egusphere-2024-3813:** Long-term nitrogen fertilization alters microbial respiration sensitivity to temperature and moisture, potentially enhancing soil carbon retention in a boreal Scots pine forest

Boris Ťupek et al.

**Reply to editor Kees Jan van Groenigen (on comments of reviewer #1 and #2)**

Your revised manuscript has now been seen by two reviewers, one of which was also involved in the first review round. Whereas the comments by reviewer #1 are easy to address, you will notice that reviewer #2 raises several substantial concerns. These need to be addressed before the manuscript can be accepted. I believe that most of these concerns can be addressed by clearly stating the limitations of your study, by better justifying some of the choices you made in your approach, and providing additional clarifications regarding your interpretation, rather than making major changes to the actual analyses. As such, I suggest minor revisions, but please note that I may contact reviewer #2 if I am not sure if the concerns have been adequately addressed.

*Dear Editor,*

*thank you for evaluating the comments of reviewers and favorable decision for minor revision.*

*We agree that the comments of reviewer #1 are easy to address and we implemented all as suggested.*

*We also agree that the comments of reviewer #2 could be considered by clarifying raised concerns in the text of our paper instead of changing the analysis. Below we reply in detail to each reviewer #2 comment and indicate corresponding text revision.*

This manuscript addresses the impact of long-term nitrogen (N) fertilization on microbial respiration and greenhouse gas (GHG) fluxes in a boreal forest ecosystem. The authors employ empirical measurements and modeling to evaluate Rh/SOC responses to temperature and soil water content (SWC), presenting this as evidence for enhanced carbon (C) retention under N addition. While the study utilizes a valuable long-term experimental site and extensive measurements, the manuscript suffers from several critical scientific, methodological, and interpretative issues that significantly undermine its novelty and reliability. Detailed recommendations for improvement are provided below.

*We appreciate reviewer #2 thoughtful comments and addressed these accordingly to substantially improve the clarity of our paper.*

Major Comments

1. The experiment lacks true replication. Only three plot pairs (CTR vs. N+) were used, separated by an average of 122 m. This design raises concerns about pseudoreplication and site effects (e.g., topography, soil heterogeneity). As the authors acknowledge (e.g., line 211),

microtopography may have affected results, yet this source of variation is not accounted for in the analysis.

*We acknowledge the reviewer's concern regarding replication and the potential influence of site-specific effects such as microtopography and soil heterogeneity. The experiment was conducted on three trenched areas per treatment (N+ and CTR), each containing two replicated 706 cm² measurement plots (i.e., 6 plot pairs in total, as clarified in line 121). These two replicate plots were treated as independent spatial observations to capture within-trench variability, while the three trenched areas per treatment were considered representative of site-level variability under each treatment.*

*We clarified the structure of the sampling design in the methods to avoid any confusion (revised text in line 121):*

**"Measurements were taken from 12 plots (six per treatment). Two plot pairs (2 × 706 cm²) were used to account for local heterogeneity in soil and microtopography at the trench level of each trenched area (1 m²), while three trenched areas per treatment were used to capture spatial heterogeneity of each treatment at the site level (Fig. 1d)."**

*However, we also fully recognize the limitations of this spatial design, and as such, we have explicitly framed the study as a case study in the conclusions (line 415), to avoid overgeneralizing the results.*

*Revised conclusions (line 415):* **Although our experiment design allowed exploratory insights into N fertilization effects, caution is needed in extrapolating beyond this site. While results represent a case study, ...**

2. Thinning treatments conducted in 1990 and 2015 differed between the control and N-fertilized plots (line 90), potentially introducing confounding effects on litter input, SOC accumulation, and soil respiration. Since thinning alters stand structure and litter quality, failing to account for these differences undermines the attribution of observed effects solely to nitrogen fertilization. A clearer separation or statistical control of thinning effects is necessary to support the current conclusions.

*We thank the reviewer for this important observation. While we acknowledge that thinning can influence soil respiration through effects on stand structure and litter inputs, previous studies suggest that these effects are generally modest (see lines 321–322). The thinning was carried out according to Finnish silvicultural guidelines (Tapio) with the aim of applying consistent intensity across all treatments.*

*We have now clarified this in the manuscript (line 95):* **"To minimize potential confounding, both CTR and N-fertilized (N+) plots were thinned in 1990 with similar intensity (~20%), and again in 2015 with nearly identical intensity, reducing basal area by**

**36.7% (CTR) and 40.1% (N+), following the Finnish silvicultural guidelines (Tapio, www.tapio.fi)."**

*Although we cannot fully rule out legacy effects of thinning, the consistent application and timing across treatments support our conclusion that N fertilization was the primary driver of the observed SOC and respiration responses.*

3. The authors use Rh/SOC to infer decomposition intensity, but Rh is inherently influenced by SOC content (i.e., autocorrelation risk). This may mask real changes in decomposition activity. The justification for SOC normalization as a proxy for microbial efficiency is weak and not supported by microbial process data (e.g., enzyme activities, community structure).

*We appreciate the reviewer's thoughtful comment regarding the potential autocorrelation between Rh and SOC, and the limitations of using $R_{h/SOC}$ as a proxy for microbial decomposition efficiency.*

*We agree that normalizing Rh by SOC does not eliminate the autocorrelation issue. However, this ratio is widely used in soil carbon studies to express decomposition intensity relative to substrate availability, especially when comparing soils with differing SOC content and is commonly applied in incubation studies to represent relative microbial activity.*

*We added a cautionary note in the Discussion to acknowledge the interpretive limitations of this approach (lines 317-324).*

**"Normalizing respiration by SOC provides a meaningful way to interpret respiration rates relative to carbon availability, especially when comparing treatments with differing SOC stocks. Although, this normalization does not fully resolve the issue of $R_h$ dependence on the amount of SOC, it has been widely adopted in field and incubation studies, as well as in soil carbon modeling frameworks (e.g., Tuomi et al., 2008; Davidson et al., 2012; Curiel Yuste et al., 2007; García-Angulo et al., 2020). Nonetheless, $R_{h/SOC}$ should be interpreted with caution, as it does not capture underlying microbial mechanisms such as enzyme kinetics or community structure, which introduce nonlinearities in the decomposition kinetics. Whereas normalizing $R_h$ by SOC as a proxy of the decomposition rate constant assumes a linear relation between decomposition rate and SOC. In the absence of microbial process data, it serves as a useful, though imperfect, indicator of decomposition rates."**

4. Although N fertilization increased Rh in absolute terms (Fig. 3a), it did not change Rh/SOC (Fig. 3b). Nevertheless, the authors claim that N enhances carbon retention (Fig. 6, lines 275–280), which lacks a consistent mechanistic link and may simply reflect dilution effects on SOC stocks.

*We appreciate the reviewer's concern regarding the interpretation of carbon retention under nitrogen (N) fertilization. While absolute Rh increased under N fertilization (Fig. 3a), Rh*

*normalized by SOC ($R_{h/SOC}$) did not increase (Fig. 3b), suggesting that the rate of $CO_2$ emission per unit of SOC remained unchanged or slightly decreased.*

*As discussed in response to Comment 3, $R_{h/SOC}$ is a widely used proxy for SOC-specific decomposition intensity. In the N-fertilized plots, increased litter input resulting from enhanced tree growth contributed to higher SOC stocks. The unchanged (or slightly lower) $R_{h/SOC}$ ratio, despite increased litter input, suggests that a smaller fraction of incoming carbon is lost as $CO_2$ - indicating more efficient carbon retention.*

*We agree that this interpretation is indirect and may partly reflect dilution effects. However, the combination of increased litter inputs and stable $R_{h/SOC}$ supports the hypothesis of enhanced SOC accumulation. We have revised lines 270–280 to more clearly present this reasoning while also noting the potential limitations and need for microbial process measurements to confirm the mechanisms involved.*

*"Annual $R_{h/SOC}$ rates (expressed as % SOC respired per year) based on daily model estimates ranged from 9.85 (±0.14 SE) to 12.95 (±0.23 SE) % and increased over 2021–2023 (Fig. 6). On average, $R_{h/SOC}$(T, SWC) values were 1.90 (±0.41 SE) % higher in control (CTR) plots compared to N-fertilized (N+) plots. This suggests that, despite increased litter inputs in N+ plots due to enhanced tree growth, the relative decomposition rate per unit SOC remained unchanged or declined, potentially favoring greater SOC retention.*

*The difference in modeled $R_h$ (calculated as $R_{h/SOC}$(T, SWC) × SOC) between CTR and N+ plots corresponds to a reduction of –345.4 (±73.6 SE) g $CO_2$ $m^{-2}$ $yr^{-1}$ in heterotrophic $CO_2$ emissions (Table 3). This potential reduction in $CO_2$ emissions outweighed the global warming potential (GWP) associated with increased $N_2O$ emissions (1.1 ± 0.1 g $CO_2$ $m^{-2}$ $yr^{-1}$ reduced $CH_4$ uptake (10.1 ± 0.5 g $CO_2$ $m^{-2}$ $yr^{-1}$) and fertilizer production emissions (6.5 g $CO_2$ $m^{-2}$ $yr^{-1}$). Overall, the net GWP balance suggests an annual reduction of -327.6 (±73.6 SE) g $CO_2$ $m^{-2}$ $yr^{-1}$ attributable to N fertilization."*

5. Equation 1 used to model Rh/SOC responses to temperature and moisture contains five parameters, which is excessive given the limited data and moderate model fit ($R^2 \approx 0.4$). In particular, key parameters for the N-fertilized treatment (e.g., parameter c) are statistically non-significant and have large standard errors (Table 1), suggesting weak parameter identifiability and potential overfitting. Therefore, the reliability of model-based annual extrapolations is questionable, and the results should be interpreted with greater caution.

*We acknowledge the reviewer's concern about model complexity and the potential risk of overfitting due to the limited dataset and moderate model fit. The five-parameter formulation was chosen to allow flexibility in capturing nonlinear and asymmetric responses of $R_{h/SOC}$ to temperature and soil moisture. Using a simpler model would have biased the fit, particularly in the control (CTR) plots where $R_{h/SOC}$ showed a clear peak response to soil water content (SWC).*

*In contrast, for the N-fertilized (N+) plots, the estimated value of the moisture response shape parameter c was close to 1 and statistically non-significant, indicating a more linear or flat response of $R_{h/SOC}$ to SWC. This aligns with the biological observation that moisture limitation was less apparent in N+ plots, possibly due to structural or microbial changes. Although the c parameter was not significant in N+, we retained the same model structure across both treatments to ensure comparability of model parameters and to enable treatment-based interpretation of response functions.*

*To reflect this limitation in interpretation, we added the following clarification to Results (lines 253-256)*

**"While not statistically significant in N+ plots, the c value near 1 suggests a relatively flat response of $R_{h/SOC}$ to SWC. In contrast, a significant c value (≈ 8, p < 0.001) in CTR plots indicates a peaked moisture response, supporting the role of moisture limitation in decomposition under ambient conditions (Table 1).**

*and in Discussion (lines 396-399):*

**"However, the lack of significance of the moisture shape parameter c in the N-fertilized treatment reflected both variability in the data and the absence of a distinct moisture optimum. The contrast between the flat response to moisture in N+ and clear peaked moisture response in CTR highlights potential treatment-related shifts in environmental sensitivity, but also underscores the need to interpret model-based extrapolations with caution."**

6. Despite a modest $R^2$ (~0.4; Table 2), the authors extrapolate hourly Rh/SOC fluxes across three years. This modeling approach is risky and not supported by comprehensive seasonal data (limited to May–October), especially given that winter fluxes are modeled without measurements.

*We agree that extrapolating $R_{h/SOC}$ fluxes year-round, particularly in the absence of winter measurements, introduces uncertainty. To address this, we have added a clarification in the Discussion about the potential limitations of annual estimates.*

*Although direct winter flux measurements were not available, the model extrapolations are based on measured temperature responses, and both treatments (CTR and N+) showed a similarly strong temperature limitation at low soil temperatures (Fig. 4). Specifically, when soil temperature fell below 5 °C, modeled $R_{h/SOC}$ values approached zero for both treatments (Fig. 5). Because these wintertime fluxes were minimal and similar between treatments, their contribution to total annual $R_{h/SOC}$ was negligible, and the impact on annual treatment differences was minor.*

*We have reformulated and added the following statement to the Discussion:*

*" Annually, N-fertilized plots respired 10.3% of SOC (± 0.3 SE), compared to 12.2% (± 0.5 SE) in CTR plots. Although the difference was derived from the modeled values, the lower respiration rate in fertilized plots suggests a potential increase in microbial carbon use efficiency, which may contribute to long-term SOC accumulation. Despite the winter fluxes not being directly measured, modeled values under low soil temperatures (<5 °C) were close to zero for both treatments due to strong temperature limitation observed in measured data. As a result, differences in winter $R_{h/SOC}$ contributed minimally to annual estimates and are unlikely to have significantly biased treatment comparisons. "*

7. The claim of net GHG mitigation (–327.6 g $CO_2$ $m^{-2}$ $yr^{-1}$; line 280) is based on a weakly supported model and fails to account for the full life-cycle impacts of fertilization (e.g., leaching, offsite emissions, biodiversity loss).

*We agree that the estimated net GHG mitigation of -327.6 g $CO_2$ $m^{-2}$ $yr^{-1}$ should be interpreted with caution. This value is based on model-derived $R_{h/SOC}$ data from a three-year period and does not fully account for all potential life-cycle impacts of fertilization, including nitrogen leaching, offsite emissions, and changes in biodiversity.*

*To reflect this, we have revised the Conclusions to include the following statement (lines 435-439):*

*"The estimated net GHG mitigation of -327.6 g $CO_2$ $m^{-2}$ $yr^{-1}$ based on $R_{h/SOC}$ model outputs from a three-year period provides a first-order approximation. However, this estimate likely underrepresents the full climate impact of fertilization, as it does not account for longer-term dynamics or potential offsite carbon and nitrogen losses, such as leaching, indirect emissions, or biodiversity-related feedback. Therefore, broader system-level assessments over longer time scales are needed to confirm these findings."*

8. Claims such as "enhanced microbial carbon use efficiency" (e.g., line 385) are not empirically validated. No microbial data (e.g., biomass, CUE assays, extracellular enzyme activity) are presented.

*We agree with the reviewer that the claim of "enhanced microbial carbon use efficiency" was not directly supported by microbial process data (e.g., biomass, CUE assays, enzyme activity). To avoid overinterpretation, we have added following sentence to discussion (line 405):*

*"However, in our study CUE could not be directly inferred from our data, as microbial process measurements were not conducted."*

9. The inference that N-induced changes in Rh/SOC response curves reflect "microbial adaptation" (line 360) is speculative without supporting microbial community or functional data.

*We removed the only claim on "microbial adaptation" (line 336).*

10. $CH_4$ and $N_2O$ fluxes are near detection limits, and treatment differences are marginal (e.g., line 200). Nevertheless, they are used to calculate $CO_2$-equivalent changes with unrealistic precision (±0.5 g $CO_2$-eq $m^{-2}$; Table 3), which exaggerates their ecological relevance.

*Thank you for the comment. As noted on line 204, the standard errors (SE) of the mean $CH_4$ (0.002 mg $CH_4$ $m^{-2}$ $h^{-1}$) and $N_2O$ fluxes (0.09 µg $N_2O$ $m^{-2}$ $h^{-1}$) used in the calculation of annual $CO_2$-equivalent SE (Table 3) are indeed smaller than the calculated detection limits for $CH_4$ (0.0238 µg $m^{-2}$ $h^{-1}$) and $N_2O$ (0.0524 µg $m^{-2}$ $h^{-1}$). This is because the SE reflects variability among replicate measurements rather than the detection limit of individual measurements. Therefore, although individual flux measurements are near detection limits, the precision of the mean flux estimates and their propagated uncertainty in $CO_2$-equivalents remains reliable and justifies the reported SE values.*

Specific Comments

1. Line 25: "Enhancing annual Rh/SOC" is contradictory if Rh/SOC is reduced annually under N+ (Fig. 6).

*To avoid any contradiction, we revised the sentence "**contrasting with a distinct humped SWC response enhancing annual $R_{h/SOC}$ in control plots**"*

2. Figure 3e–f: Soil temperature and moisture differ only slightly between plots, yet these small differences are used to explain large GHG flux differences. This is overstated.

*We agree that the overall differences in soil temperature and moisture between plots are small. However, it is not the magnitude of these differences alone that explains the GHG flux variation, but rather the differing sensitivities and dependencies of soil respiration to soil moisture and temperature under different treatments.*

*We clarified this point in the manuscript to avoid overstating the role of small absolute differences in environmental variables.*

3. Figures 5–6: These figures show model outputs rather than direct observations and should not be interpreted as empirical findings without caution.

*We revised the manuscript (e.g., revised chapter 4.3) to clearly distinguish between model outputs and empirical measurements, emphasizing the model-based nature of simulated lines in Figures 5a and annual sums in Figure 5b and the associated uncertainties.*

4. Line 300: "Reduced microbial respiration" contradicts earlier statements that Rh increased in fertilized plots.

*Thank you for pointing this out. To clarify, the phrase "reduced microbial respiration" in line 300 was intended to refer specifically to a reduction in the $R_{h/SOC}$ ratio, rather than an absolute*

*decrease in $R_h$. We corrected this in added text (line 324).* **"Although, results from our study suggested a reduction in the $R_{h/soc}$ ratio, rather than an absolute decrease in $R_h$."**

5. Stating that the models showed "relatively large residuals" but "mean bias errors were close to zero" is not a valid justification for model adequacy.

*We agree that simply stating "mean bias errors close to zero" is insufficient to justify model adequacy. Although our empirical models used to evaluate the sensitivity of soil respiration to soil temperature and moisture showed a moderate coefficient of determination ($R^2 \approx 0.4$), other performance metrics - such as root mean square error (RMSE), mean absolute error (MAE), and mean bias error (MBE) - were relatively low, suggesting that the models captured overall trends with acceptable accuracy.*

*However,* **we removed these details in the conclusions to focus on the actual results**

6. All tables should be formatted using the standard three-line table format.

*Thank you for this formatting reminder which we now follow accordingly.*

---

## Author Response (AR3)

**Egusphere-2024-3813:** Long-term nitrogen fertilization alters microbial respiration sensitivity to temperature and moisture, potentially enhancing soil carbon retention in a boreal Scots pine forest

Boris Ťupek et al.

**Reply to editor Kees Jan van Groenigen (on technical comment)**

Dear authors, thank you for sending this revised version. As the concerns raised by both reviewers have been addressed, this manuscript can now be accepted for publication. I recommend "technical corrections", because one line in the newly added text could use some adjustments:

"The estimated net GHG mitigation of -327.6 g $CO_2$ m-2 yr-1 based on Rh/SOC model outputs from a three-year period provides a first-order approximation. However, this estimate likely underrepresents the full climate impact of fertilization, as it does not account for longer-term dynamics or potential offsite carbon and nitrogen losses, such as leaching, indirect emissions, or biodiversity-related feedback. Therefore, broader system-level assessments over longer time scales are needed to confirm these findings."

"However, this estimate.... feedback." is problematic, as "underrepresents" suggests that the GHG mitigation due to N addition is underestimated in this study. The opposite seems much more likely; all the processes you list would decrease the climate benefit. As such, I suggest you change "likely underrepresents the full climate impact" to "This estimate does not fully capture the climate impact of fertilization, as...".

May I also ask you to take into account the notification from the review file validation process? Please place sections "Acknowledgements", "Data and code availability", "Author contribution" and "Competing interests" before the reference list. The manuscript structure should follow the guildelines as described at https://www.biogeosciences.net/submission.html#manuscriptcomposition

all the best,

Kees Jan van Groenigen

*Dear Editor,*

*thank you for reviewing the revised manuscript and for your insightful suggestion regarding the phrasing of N fertilization impacts.*

*We agree with the comment and reformulate the sentence (L415):*

***"However, this estimate does not fully capture the climate impact of fertilization, as it does not account for longer-term dynamics or potential offsite C and N losses, such as leaching, indirect emissions, or biodiversity-related feedback.***

*We also placed sections "Acknowledgements", "Data and code availability", "Author contribution" and "Competing interests" before the reference list.*

*Kind regards,*

*Boris Tupek*